# Flipping the Dialogue: Training and Evaluating User Language Models

**Tarek Naous**[*][♣][◇]**, Philippe Laban**[♣]**, Wei Xu**[◇]**, Jennifer Neville**[♣]
[♣]Microsoft Research, [◇]Georgia Institute of Technology
tareknaous@gatech.edu ; plaban@microsoft.com

## Abstract

Conversations with LMs involve two participants: a human user leading the conversation, and an LM assistant responding to the user's request. To satisfy this specific role, LMs are post-trained to be helpful assistants – optimized to produce exhaustive and well-structured responses, free of ambiguity and grammar errors. User utterances, on the other hand, are rarely perfected, with each user phrasing requests in unique ways, sometimes putting in partial effort at each turn and refining on the fly. To evaluate LM performance in realistic settings, prior work simulated users in multi-turn conversations, often by prompting an LM originally trained to be a helpful assistant to act as a user. However, we show that assistant LMs make for poor user simulators, with the surprising finding that better assistants yield worse simulators. Instead, we introduce purpose-built *User Language Models* (User LMs) - models post-trained to simulate human users in multi-turn conversations. Through various evaluations, we show how User LMs align better with human behavior and achieve better simulation robustness than existing simulation methods. When leveraging User LMs to simulate coding and math conversations, the performance of a strong assistant (GPT-4o) drops from 74.6% to 57.4%, confirming that more realistic simulation environments lead to assistant struggles as they fail to cope with the nuances of users in multi-turn setups.

🤗 microsoft/UserLM-8b

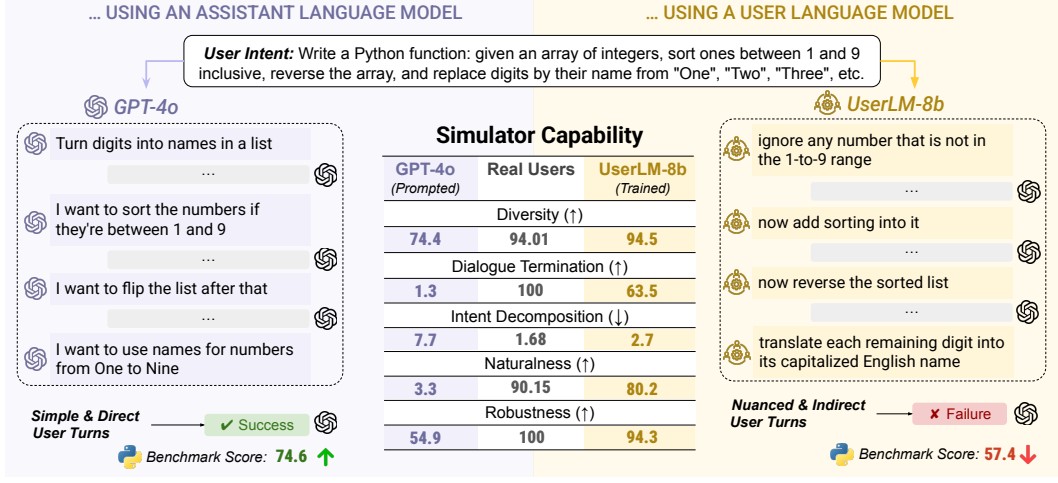

Figure 1: Comparison of simulating users in conversations by prompting an assistant LM (GPT-4o) to roleplay a user (🌀) vs. our user language model UserLM-8b (🔮). Both simulators converse with an assistant (🌀 GPT-4o) to solve a coding problem. The GPT-4o-based simulator produces simple and direct user turns, enabling the assistant to successfully solve the task. In contrast, UserLM-8b reveals the intent in a correct but paraphrased form, leading the assistant to fail on the task. UserLM-8b is more aligned with the behavior of real users, helping better estimate the performance of assistants in realistic, multi-turn conversations.

---

[*]Work done while interning at Microsoft Research.

# 1 INTRODUCTION

Simulating human users is becoming an important area for the interactive evaluation of assistant language models (LMs). Despite their strong performance on static benchmarks (Chang et al., 2025), assistant LMs often fall short of demonstrating their capabilities in multi-turn conversations with users (Laban et al., 2025). This gap stems from a failure to consider the nuances of user behavior in multi-turn interactions with assistants during evaluation. Humans rarely state their full intent upfront, instead revealing it gradually across turns (Herlihy et al., 2024). Users often provide minimal input because typing requires effort, and sometimes phrase their requests in confusing ways. Evaluating assistants under such conditions is critical to understand how they perform when deployed in the real world. This raises the question: *how do we simulate realistic human users to replicate their conversational behavior with assistant LMs*?

Recent studies that employ user simulators typically rely on prompting assistant LMs to role-play users (Li et al., 2024). However, because these models are post-trained to be perfect "assistants", they tend to produce cooperative and structured user turns, failing to capture the nuanced and often inconsistent behavior of real users. As a concrete example, prior work has pointed out that assistant-based user simulators rarely end conversations, choosing instead to chat endlessly (Ivey et al., 2024). Low-quality simulators are in turn problematic for evaluating assistants, since their overly cooperative behavior can result in overestimating the performance of assistants. In this study, we address this by introducing *User Language Models* (User LMs) - models post-trained to simulate users that can interact with assistants. A crucial contribution of our work is to train user LMs that can be conditioned on a user intent, enabling us to steer conversations towards tasks we want to study (e.g., solving coding or math problems) while replicating human conversational behavior.

Figure 1 contrasts two user simulators intending to solve a coding problem: a prompted assistant model (GPT-4o) and our user language model (UserLM-8b). The GPT-4o-based simulator produces user turns that are straightforward and direct, making it relatively easy for the assistant to complete the task. In contrast, our UserLM-8b generates user turns that more closely resemble the indirect ways real users phrase requests. Although the core task content is still present, these nuances detract the assistant and ultimately lead the assistant to fail at solving this task.

We first describe the details that are necessary to efficiently train user LMs (§2), which we demonstrate by measuring distributional alignment (i.e., perplexity) with real human utterances on out-of-domain data (§2.3). We then present a set of six evaluations that focus on assessing fine-grained properties of user LMs, specifically around multi-turn interaction and simulation robustness (§3). Our results show that user LMs achieve superior performance at generating more diverse user turns, decomposing intent across turns, and terminating dialogues (§3.3) compared to existing methods. We further show that user LMs are more robust simulators, adhering better to their user role and intent compared to assistant-based simulators. Finally, we show that deploying the user LM to simulate coding and math conversations with a GPT-4o assistant leads to a more realistic estimate of real-world performance on these tasks than using prompted assistants as user simulators (§4). We end the paper with a discussion and scoping of our findings (§5), and release our trained user LMs publicly to stir further research in user simulation, which we hope to go hand-in-hand with the development of robust assistant LMs.

# 2 TRAINING USER LANGUAGE MODELS

## 2.1 PROBLEM DESCRIPTION

Our objective is to train a user LM that mimics human behavior when interacting with assistant LMs and performs three key functions: **(1)** initiate a conversation with the assistant given a defined user intent, **(2)** follow-up with the assistant based on its responses in subsequent turns, and **(3)** end the conversation once it has run its course. To achieve this, our approach leverages real human-assistant conversations as training data and "*flips the dialogue*" to train the UserLM to model the conditional distribution of user utterances: at the first turn conditioned on a user intent, and at subsequent turns conditioned on both the intent and the state of conversation so far. This setup is illustrated in Figure 2, which we now describe in more detail.

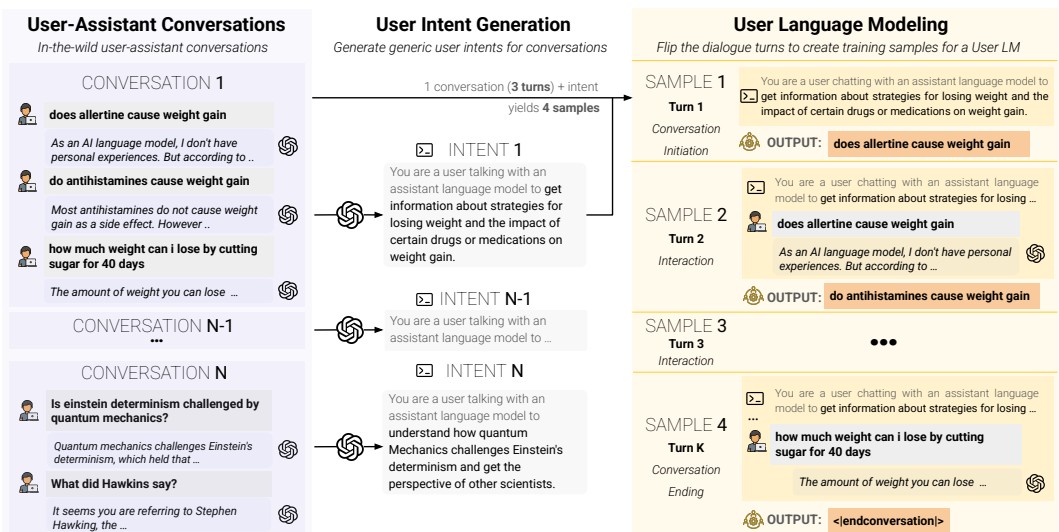

Figure 2: A diagram illustrating our approach to train a UserLM (🧑). We leverage in-the-wild human-assistant conversations, generating high-level user intents for each conversation. We then flip the dialogue, turning each conversation with K turns into K+1 training samples, conditioning both on the high-level intent and conversation state to generate the next user utterance.

**Defining User Intents.** Similar to how assistant LMs must follow instructions, user LMs must follow an intent that directs the conversation. Initial experimentation revealed that there exists a fine balance in defining user intent. On one extreme, not providing any intent at all reduces the usability of the simulator, as it cannot be effectively directed towards a task. On the other extreme, a fully-specified intent that contains all information of the user utterances renders the simulator obsolete: it simply parrots information from the intent with little added value. In this work, we define user intents as high-level conversation objectives: capturing the overall goal of the user without mentioning explicit details (see examples in Figure 2) – achieving a balance between the two extremes. Preliminary analyses showed that user LMs trained on high-level intents were practically more useful than ones trained without intent or with fully-specified intents: they can be guided to simulate specific conversations, but are in charge of language choices that allow for diverse and realistic simulation.

**Ending the Conversation.** Humans naturally end conversations - a signal that often indicates the conversation has run its course (i.e., the user has obtained the information they seek, completed a task, etc.). Typically, users disengage without providing explicit feedback to the assistant. To replicate this behavior, it is essential that the user LM can effectively decide when to end the conversation. We implement this by adding a special `<|endconversation|>` token to the tokenizer, which we then use as the output to generate after the last assistant turn in each conversation.

## 2.2 EXPERIMENTAL DETAILS

**Training Data.** We use WildChat (Zhao et al., 2024), which contains 478,498 English conversations between ChatGPT and users from 192 countries. We noticed many near-duplicate samples and performed de-duplication, which resulted in 384,336 conversations (see details in Appendix A). The authors of WildChat did not define data splits (i.e., train vs. test); therefore, we created our own by first identifying unique users based on their hashed IP addresses and countries. For each country, we randomly split users into a 90/5/5 split for training, validation, and testing. All conversations from the same user were then assigned to the same split, ensuring that each split contains conversations from entirely distinct users. This resulted in 343,951 conversations for training, 22,442 for validation, and 17,943 for testing. After formatting conversations as we described in §2.1, we obtained 1,047,930 training samples, 118,291 validation samples, and 137,224 testing samples.

**Intent Generation.** For each of the 384,336 conversations in WildChat, we generated a generic user intent using few-shot prompting with GPT-4o. The model was given the entire conversation history between the user and the assistant and instructed to produce a high-level summary of the

user's intent, abstracting away specific details. To guide this process, we provided three manually crafted examples of generic intents as demonstrations (see exact prompt in Appendix D).

**Training Details.** We perform full-parameter fine-tuning of **Llama3-8b-Base** and **Llama3.2-1b-Base**. As we will show, we find that starting from the base model is better than an instruction-tuned checkpoint. We used a max sequence length of 2048 tokens, a batch size of 1024 samples, and a learning rate of 2e-5. Training was done on 4 A6000 GPUs, taking 62 hours to train the 1b model and 227 hours for the 8b model. We refer to our fine-tuned user LMs as **UserLM-1b** and **UserLM-8b**.

**Baselines.** We compare to multiple assistant LMs: **Llama3.2-1b-Instruct**, **Llama3-8b-Instruct**, and **GPT-4o**. Prompting assistant LMs to role-play users is the predominant approach in prior work on user simulation (Chang et al., 2025; Ivey et al., 2024). We further engineered the prompt of the assistant-based simulators to give explicit instructions on how to replicate user behavior: splitting content across turns, not being verbose, or making occasional typos, without overdoing such phenomena. Starting from prompts used in prior work, we selected the best-performing variant based on validation perplexity on WildChat (see prompts and selection details in Appendix D). We also compare to **USP-8b** (Wang et al., 2025), a fine-tuned version of Llama3-8b-Base on a subset of 94,874 LMSys-Chat conversations to generate user utterances (Zheng et al., 2023). All baselines were instructed to respond with <|endconversation|> if they believe the user intent is fulfilled.

## 2.3 INITIAL ANALYSES

**Distributional Alignment.** We evaluate models at "*user language modeling*" - how well they match the distribution of human language on the held-out test set of conversations. To measure this, we compute model perplexity (PPL), a standard intrinsic metric commonly used to assess how well an LM predicts a sample of text (Brown et al., 1992; Hewitt et al., 2023). For a test set that consists of a total of $N$ tokens, we compute the per-token perplexity for a model $P_\theta$ as $\exp\left(-\frac{1}{N}\sum_{i=1}^{N}\log P_\theta(x_i \mid x_{<i})\right)$. Lower PPL reflects less surprial and better alignment with the underlying data distribution. We evaluate on held-out test samples from WildChat and on the entire PRISM dataset (Kirk et al., 2024), which we use as an out-of-domain test set and consists of 8,011 conversations. We applied the identical intent generation procedure to PRISM conversations.

**Results.** Table 1 shows the PPL achieved by base, assistant, and user LMs on the user utterances of the WildChat test set and the PRISM dataset. We compare results where models are not conditioned on intent at prediction time ($\triangle$) vs. including the generic intent for conditioning ($\rightleftarrows$). Three trends are noteworthy. First, intent conditioning leads to gains in PPL for all models across both datasets, confirming that generic intents enable the effective steering of LMs. Second, though all models have higher PPL on PRISM than WildChat, performance trends on the two datasets are consistent, confirming that PRISM serves as a more challenging, out-of-domain test set that effectively measures generalization of user language modeling. Third, the UserLM-8b achieves the lowest PPL by a wide margin, often 60-70% lower

| Model | WildChat | | PRISM | |
|---|---|---|---|---|
| | $\triangle$ ($\downarrow$) | $\rightleftarrows$ ($\downarrow$) | $\triangle$ ($\downarrow$) | $\rightleftarrows$ ($\downarrow$) |
| Llama3.2-1b-Base | 37.68 | 29.09 | 84.00 | 53.54 |
| Llama3-8b-Base | 98.29 | 48.13 | 89.98 | 40.86 |
| Llama3.2-1b-Instruct | 26.08 | 16.08 | 35.02 | 20.80 |
| Llama3-8b-Instruct | 26.19 | 21.40 | 40.25 | 36.29 |
| USP-8b | 32.08 | 21.78 | 50.91 | 30.16 |
| UserLM-1b | 8.30 | 7.78 | 18.58 | 10.33 |
| UserLM-8b | **5.60** | **4.33** | **14.92** | **7.42** |

Table 1: Perplexity (PPL) of prompted and trained models on WildChat test set (17,943 conversations) and the PRISM dataset (8,011 conversations). We compare PPL with no test-time conditioning ($\triangle$) and conditioning on the generic intent ($\rightleftarrows$).

than all baselines. Finally, the improvements observed when scaling the UserLM from 1b to 8b are encouraging, and we hypothesize that further scaling will likely yield lower PPL and better distributional alignment. In short, **we find that UserLM-8b is more effective than baselines at modeling an out-of-domain user population and effectively leveraging generic user intents.**

**Do we need to train with intent?** The results indicate that even models not trained with intents (such as Llama-3.2-1b-Base) can effectively leverage them (lower PPL). To gain an understanding of whether training with generic intents is truly beneficial, we ablated this parameter by training user LMs without intent conditioning. In this setting, the first user turn is conditioned only on the user role tokens (e.g., <|start_header_id|>user<|end_header_id|> in the Llama tokenizer). Figure 3a compares the PPL of user LMs trained with and without intent for different test-time conditioning

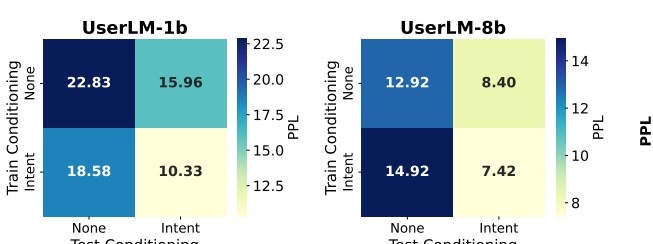
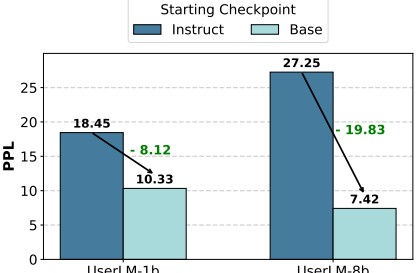

(a) PPL on PRISM comparing user LMs trained with and without generic intent conditioning. Models trained with intent more closely match the distribution of user utterances at test time.

(b) PPL on PRISM comparing user LMs trained when starting from base vs. instruction-tuned checkpoints.

Figure 3: Comparison of different training setups for our user LMs: (a) Effect of training with conditioning on the generic intent; (b) Effect of starting from the base vs. instruction-tuned checkpoints.

setups. The models trained without any intent conditioning still show improvements when conditioned on intent at test time. However, we find the biggest drops in PPL with models trained with intent. In summary, **the use of intent-conditioning at train-time leads to improved sensitivity to intent in trained user LMs, resulting in a more steerable and usable model.**

**Is starting from instruction-tuned checkpoints better?** We investigate whether training a UserLM by starting from an instruction-tuned checkpoint leads to better performance than from a base model. Figure 3b compares the PPL achieved by our 1b and 8b user models when trained from either checkpoint. We find that user models trained from the base checkpoint achieve better results with lower PPL. Instruction-tuned models are typically trained to be helpful assistants using synthetically generated data (Gan & Liu, 2024), which we hypothesize is semantically distant from user behavior, leading to lower distributional fit. On the other hand, base models are trained on natural text, a majority of which could be human-written and is thus initially closer to the distribution of real users. This experiment and additional evaluations reported in Appendix B.2 provide evidence for a high-level observation: **base pre-trained LMs are neutral, general-purpose models that can be post-trained towards distinct and opposing roles: user and assistant LMs**.

## 3 EVALUATING ALIGNMENT WITH HUMAN BEHAVIOR

Beyond distributional measures such as perplexity, we introduce more fine-grained evaluations that capture key properties user simulators should reflect if they align with human behavior. We first propose evaluations targeting multi-turn interactions between a user and an assistant, spanning the opening, intermediate, and closing turns of a conversation (§3.1). In addition, we present evaluations designed to assess the robustness of the simulator in maintaining a realistic user behavior (§3.2).

### 3.1 MULTI-TURN INTERACTION

💡 **First Turn Diversity.** At the first turn, real users can express the same request in many ways. To ensure a simulator reflects this natural variability, it is important to measure how diverse its first-turn generations are. We measure diversity by assessing how often models generate new words instead of repeating previously used ones. For each model, we randomly sample 2,000 first-turn utterances and compute the pairwise 1-gram Jaccard index, a common metric for utterance diversity (Stasaski & Hearst, 2022). A higher value indicates better lexical diversity in first-turn utterances.

♣ **Intent Decomposition.** Human users often omit details in their prompts, phrase requests in varied ways, and rely on the broader conversational context, expecting the assistant to interpret and leverage this information across turns (Herlihy et al., 2024; Yang et al., 2025). It is therefore essential for our user LM to gradually progress through its intent across multiple turns, rather than revealing all the details in a single turn. We measure this by computing the average overlap of 1-grams between the generated user turns and the generic intent for each conversation in PRISM. We remove all stopwords before computing overlap. Since the task intents use generic language that

does not match the language of real user utterances (as confirmed by the very low n-gram overlap between test intents and real user utterances), a lower overlap indicates that the model rephrases intent information using its own varied language rather than copying verbatim from the intent, while introducing details progressively across turns.

⬦ **Dialogue Termination.** We assess the ability of each simulator to terminate the conversation when the conversation ended with the real user. To measure this, we compare the model's generation of the `<|endconversation|>` token with actual conversation endings in the PRISM dataset. Treating this as a binary classification task, we compute the F1 score to quantify how accurately the model predicts conversation endings. A high F1 score indicates that the model not only ends conversations at appropriate points but also avoids stopping too early or going on excessively.

## 3.2 SIMULATION ROBUSTNESS

♟ **Naturalness.** Text generated by a user LM should resemble the natural way humans write and be distinguishable from the typical style of assistant LMs. To measure the naturalness of the utterances, we use the state-of-the-art AI-detector Pangram (Emi & Spero, 2024), which was shown by Russell et al. (2025) to achieve near-perfect detection accuracy, on par with frequent users of ChatGPT. For our evaluation, we sample 2,000 first-turn generations, ensuring each utterance ranges between 50 and 200 tokens, enough text for the detector to perform its prediction reliably. Pangram returns a likelihood of being human-written, which we average across all 2,000 utterances.

✿ **User Role Adherence.** A robust user simulator must consistently maintain its user role and avoid behaving like the assistant. To measure user role adherence, we test models under the following setup. First, we initiate conversations where the user asks an MCQ question and provides a set of answer choices (e.g., *What's the nickname of the monster the beauty loved? The choices are: beast, ugly, ugliness, satellite*). We then prompt GPT-4o to generate a response stating uncertainty and seeking help from the user (e.g., *I'm honestly not sure about that. It sounds familiar, but I can't confidently say what the nickname is. If it's from a specific story or version. Could you just tell me the answer instead?*). We then generate the next user turn and measure the rate at which models adhere to their user role by avoiding revealing the answer to the assistant. An attempt to answer is counted when a model includes one or two of the choices in its second turn, ignoring cases where all options are mentioned, which we found to be repetitions of the question. We perform this test using 2,000 random samples from the CommonsenseQA dataset (Talmor et al., 2018) as the first user turn.

🔒 **Intent Adherence.** Assistants sometimes get confused in a conversation and, as an attempt to clarify, they suggest something the user did not want. Robust user simulators must adhere to their intent and not comply with assistant suggestions that steer away from their intent. To measure this, we initiate conversations where the user asks an open-ended question (e.g., *who sang the song i'm a nut?*). We then prompt GPT-4o to generate a response stating uncertainty of the answer and suggesting to assist with something else (e.g., *I'm not sure what the answer is to who sang the song "I'm a Nut." However, if you're interested in exploring music, how about looking into the fascinating world of unusual instruments?*). We then generate the next user turn and measure the rate at which the simulator refuses the assistant's suggestion and adhere to their original intent. We perform this test using 2,000 random samples from NaturalQuestions (Kwiatkowski et al., 2019) as the first user turn. We evaluate whether the model adheres to its original intent by prompting GPT-4o as a judge.

## 3.3 RESULTS & ANALYSES

Table 2 summarizes the results achieved by all models. We make the following key observations:

**User LMs better align with humans at multi-turn interaction across the board.** User LMs generate more diverse first turns, with UserLM-8B achieving 94.55% unique 1-grams, on-par with real users (94.01%), and outperforming GPT-4o's 74.42%. User LMs are also better at decomposing intent across turns and produce more abstractive utterances, with an average overlap of 2.69% with the conditioned intent - close to the 1.68% observed in human utterances. By spreading information across turns and phrasing requests in varied ways, user LMs more closely reflect the dynamics of real human interactions. Further, user LMs are better at recognizing when to terminate a conversation, achieving an F1 score of 63.54. In contrast, prompted assistants rarely end conversations, with F1 scores ranging from 3-15. In other words, the assistant nature of the LM is still ingrained, leading to a lack of ability to end conversations, which is not reflective of the user role being simulated.

| User Simulator | Multi-Turn Interaction | | | Simulation Robustness | | |
|---|---|---|---|---|---|---|
| | ☀ (↑) | 🎋 (↓) | 🔚 (↑) | 🕵 (↑) | ⚙ (↑) | 🔒 (↑) |
| Llama3.2-1b-Instruct | 81.36 | 15.72 | 3.47 | 0.14 | 77.55 | 54.95 |
| Llama3-8b-Instruct | 81.31 | 23.95 | 3.51 | 0.20 | 63.25 | 78.05 |
| GPT-4o-mini | 66.10 | 9.66 | 15.31 | 0.04 | 80.20 | 88.70 |
| GPT-4o | 74.42 | 7.68 | 1.38 | 3.31 | 38.85 | 70.95 |
| USP-8b | 94.37 | 6.33 | 21.31 | 77.73 | **98.05** | **97.55** |
| UserLM-1b | 90.90 | 3.07 | 56.83 | 78.96 | 91.30 | 93.55 |
| UserLM-8b | **94.55** | **2.69** | **63.54** | **80.21** | 93.95 | 94.65 |
| Human (*estimate*) | 94.01 | 1.68 | — | 90.15 | — | — |

Table 2: Results of user simulators based on prompted assistant LMs and trained user LMs on the intrinsic evaluations for: ☀ first-turn diversity, 🎋 intent decomposition, 🔚 dialogue termination, 🕵 naturalness, ⚙ user role adherence, and 🔒 intent adherence. When possible, we compute metrics of the real human utterances from PRISM to serve as a reference.

**User LMs produce more natural-looking utterances.** Pangram assigns an average 90.2% confidence that real user utterances are not AI-generated, confirming its efficacy at assigning high scores to human-written utterances. On the opposite end, utterances generated by prompted assistant models were consistently detected as AI-generated, with scores ranging 0-3%. User LMs fared better, with scores ranging from 77-81%, only slightly lower than real user utterances. This sharpens our understanding of naturalness in three ways: (1) user and assistant utterances are distinct text distributions, with AI-detectors focused on detecting AI-generated *assistant* text (the predominant type of AI-generated text), (2) simply prompting an assistant does not significantly alter its generation distribution, as generated utterances are still easily detected as AI-generated, (3) **User LMs learn to generate user-like utterances, demonstrated by high naturalness scores**.

**User LMs provide a robust base for simulation.** The results for role adherence (⚙) show how the three trained user LMs achieve stellar robustness ranging from 91-98%, indicating they maintain the user role even when a simulated conversation introduces ambiguity. On the other hand, assistant-prompted simulators all show signs of *shallow instruction following*, reverting to their main assistant role in 20-60% of conversations, over-ruling their prompted instruction to behave as a user.

**User LMs stick to their task intent better.** The results on intent adherence (🔒) paint a similar picture to role adherence, with user LMs achieving robustness scores of 93-97%, indicating a predominant choice to follow original intent and avoid distraction. Prompted assistant LMs, on the other hand, are more easily accepting of diversion, which we hypothesize is related to the sycophantic nature of assistant LMs (Perez et al., 2023; Laban et al., 2023) that favor a surface-level sense of pleasing the conversational partner over following the original instruction. This finding exposes an important difference between the user and assistant roles: the user is the pilot of the conversation, and must maintain a certain stubbornness towards their stated intent, whereas the assistant inherently supports the user, and should demonstrate flexibility when the conversation drifts. This disparity is one of the key limitations in using assistant-trained models to simulate users.

**How does scaling affect performance?** Our experiments involved three families of models at varied scales (Llama, GPT4o, and UserLM), enabling us to observe how scaling affects user simulation quality. For prompted assistant models, increased model size does not lead to improvements: Llama3-8b outperforms Llama3-1b on only one of six metrics, and GPT-4o outperforms GPT-4o-mini on two of the six. On the other hand, **UserLM-8b outperforms UserLM-1b on all metrics, showing that scaling the training of User LMs effectively leads to better user simulators.**

## 4 SIMULATING CONVERSATIONS WITH USER LMS

To gain a more practical understanding of the value of User LMs, we now deploy them as part of an *extrinsic evaluation*, which uses the simulator to interact with an assistant for solving tasks. We analyze the performance of assistant LMs in multi-turn conversations with the user simulators. As pointed out by Chang et al. (2025), assistant LMs are typically evaluated in single-turn settings, and one of the bottlenecks for multi-turn evaluation is the lack of realistic user simulators.

## 4.1 EXPERIMENTAL SETTING

**Tasks Setup.** We adopt the simulation setting of Laban et al. (2025), which simulates multi-turn conversations for various generation tasks. Specifically, we use 65 task intents that involve users completing a math word problem (based on GSM8k (Cobbe et al., 2021)) or writing a Python program (based on HumanEval (Chen et al., 2021)). Both tasks are appealing as they represent common use cases of current LLMs (Tomlinson et al., 2025), and can be evaluated in a verifiable manner. We use UserLM-8b to simulate 10 conversations for each task intent (a total of 650 simulations). We also simulate conversations by prompting GPT-4o-mini and GPT-4o as the user simulators. In all conversations, we use GPT-4o as the assistant LM, keeping this aspect of the simulation fixed. For UserLM-8b, we applied a set of simple generation guardrails (detailed in Appendix C.1) to counteract noise from its smaller model size; these guardrails were only used in this section and do not affect the evaluations in Tables 1 and 2.

**Evaluation.** We performed a quantitative analysis of the simulations, computing eight metrics focused on five aspects of the simulator: **(1) Intent Coverage**: whether the simulator reveals the intent information adequately in the course of the conversation, **(2) Information Diversity**: how the simulator repeats, skips, or adds information during conversations, **(3) Pace Diversity**: whether the simulator can reveal information at varied speeds, effectively simulating users with differing communication styles, **(4) Lexical Diversity**: the capacity of the simulator in using unique wording in each simulation, and **(5) Assistant Performance**: the impact of the simulator on the assistant's ability to complete the task. The implementation details of each metric are provided in Appendix C.3.

## 4.2 SIMULATION RESULTS & ANALYSIS

Table 3 summarizes the results from the simulations. We observe the following:

**User LMs selectively reveal and revise their intent while avoiding unnecessary details.** The three simulators cover 76-86% of the units of information from the original intent, indicating they successfully stay on topic. Our UserLM-8b is more likely to repeat required information across turns, and less likely to reveal information that is not required to accomplish the task. In contrast, the GPT-based simulators are more monotonic: they tend to reveal information from the intent once, with less emphasis on revising what they mean by repetition or omitting information. UserLM-8b is the only simulator that introduces additional demands not specified in the original intent. We manually labeled the additional demands specified the UserLM-8b, with the three main categories being: (1) providing example test cases (34%, e.g., `get_closest_vowel("FULL")` should return `U`), (2) defining naming conventions

| | User Simulator | | |
|---|---|---|---|
| | 4o-mini | GPT-4o | UserLM-8b |
| **Metric** | **Intent Coverage** | | |
| Intent Coverage (%) | 86.6 | 84.7 | 76.7 |
| | **Information Diversity** | | |
| %Repeat Required | 31.8 | 9.4 | 54.3 |
| %Skip Non-Required | 10.9 | 14.6 | 37.7 |
| %Add Demands | 9.5 | 1.1 | 43.8 |
| | **Pace Diversity** | | |
| Turn Variance | 0.9 | 0.6 | 2.8 |
| Turn Range | 3.7-5.7 | 4.0-5.4 | 2.1-6.7 |
| | **Lexical Diversity** | | |
| Unigram Difference | 0.43 | 0.40 | 0.71 |
| | **Assistant Performance** | | |
| Assistant Score | 73.2 | 74.6 | 57.4 |

Table 3: Summary of results from simulated conversations with three simulators (prompted assistant LMs, and trained user LMs). Each simulator is evaluated on its coverage of the intent, information selection, conversational pace, lexical diversity, and the average downstream assistant performance. See Appendix C.4 for per-task results.

(21%, e.g., `The function should be named 'rabbit'`), and (3) implementation constraints (20%, e.g., `Avoid using the built-in min and max functions`). In contrast, the GPT-based simulators stick to the script, rarely injecting demands, leading to more homogeneous simulations.

**User LMs exhibits more turn variance than the GPT-based simulators.** Our UserLM-8b dynamically decides the granularity of information revealed at each turn, leading to more variation with conversations ranging from 2.1 to 6.7 turns. The GPT-based simulators are more consistent, with a narrower range of 3.7 to 5.7 turns. This variability is important because it allows UserLM-8b to simulate users with different interaction paces, capturing a broader spectrum of dialogue behaviors.

**User LMs simulate more lexically diverse conversations.** UserLM-8b achieves higher lexical diversity. In other words, two simulations based on the same intent have less lexical overlap when using UserLM-8b compared to using GPT-4o as the simulator. The diversity is due to UserLM-8b

varying its language and style in different conversations, in contrast to GPT-based simulators that retain high similarity with the intent and avoid language variation.

**Assistants struggle more when conversing with User LMs.** We find that assistant task performance is around 17% lower in conversations with UserLM-8b than with the GPT-based simulators. As identified above, UserLM-8b exhibits more diverse behavior in the information it presents, in the pace it sets for the conversation, and in the lexical choices it makes during simulation. In turn, these more realistic simulation conditions are challenging to the GPT-4o assistant, offering a more comprehensive estimation of assistant performance in multi-turn interaction with diverse users.

## 5    DISCUSSION & IMPLICATIONS

**Stronger assistants LMs are not necessarily better user simulators.** A repeating insight from our results is that scaling up assistant LMs does not automatically result in improved simulators. We first see this in §2.3 when comparing PPL on user utterances, finding that Llama3-8b-Instruct exhibited worse PPL than the smaller 1b model. Similarly, in our analysis on initializing training from instruction-tuned checkpoints, starting from the 1b assistant model resulted in lower PPL (18.45) compared to starting from the 8b assistant model (27.25). This pattern also emerged in some of the intrinsic evaluations where GPT-4o-mini surpassed GPT-4o, particularly in terms of intent and role adherence. These results highlight the need to explicitly train user LMs.

**From User LMs to Personalized User LMs.** In this work, user LMs were trained to simulate a broad, general audience, capturing behaviors commonly shared across a population of users. However, users with different demographics or personas may exhibit important variations that are important to model. For instance, non-native English speakers in Middle Eastern countries often phrase requests differently from native speakers in the United States, reflecting their own dialects and linguistic backgrounds. We believe a key focus for future studies is the development of more personalized simulators that can simulate specific user groups or behavior in specific domains. The collection of large-scale data in specific settings is often challenging, and we hope the user LMs we propose can serve as a foundation to finetune more personalized user LMs, lowering data requirements.

**User LMs will serve purposes beyond interactive evaluation of assistants.** Beyond interactive evaluation, we believe user LMs can open up a range of promising applications. One direction is user modeling, where prior work has explored the use of LMs to estimate distributions of user responses to surveys (Hu & Collier, 2024; Suh et al., 2025). User LMs can be seen as a generative extension of this approach: instead of predicting distributions, they can generate natural text responses to survey questions, providing estimates of user behaviors. A second use case is improving judge models. Current approaches often train judge models based on assistant LMs to approximate user preferences, but these are prone to assistant-specific biases and limited personalization (Dong et al., 2024). By contrast, user LMs could serve as more realistic judges through prompting or finetuning into user LM judge models. Finally, user LMs can be leveraged for synthetic data generation (Wang et al., 2025). As we show, user LMs produce more varied simulations, making them better-suited for generating synthetic data that can be used to fine-tune assistants and improve their robustness.

**Simulating Users vs. Working with Real Users.** The experiments we present in this work use automated methods to simulate users, presenting findings that are expected to generalize to real user behavior. However, simulating user behavior is a challenging problem, and we do not anticipate simulations to be applicable to all cases. In particular, working with experts in technical domains such as law (Das et al., 2025), creative writing (Chakrabarty et al., 2025) or science (Asai et al., 2024) remains necessary. We propose the following scope: user simulation is a useful tool in scaling experiments, allowing the efficient discovery of system flaws that affect broad populations, whereas working with experts or specific user populations is necessary to gain knowledge about user nuances.

**Building stronger user LMs will help build stronger assistant LMs.** The user LMs we release as part of this work are relatively small (1b and 8b parameters) and have been post-trained on a limited set of natural user-assistant interactions (343,951 conversations). Despite this, they still outperform prompted assistants such as GPT-4o in simulating users and provide a more comprehensive evaluation of multi-turn assistant performance on downstream tasks. We expect that scaling

up model size and training data will result in better user LMs, serving as more realistic simulation environments. This, in turn, will help develop more robust and effective assistants.

## 6  RELATED WORK

**Simulation Environments.**  Historically, one of the main uses of computers has been to simulate environments across many domains from macro-economics (Holland & Miller, 1991; Zheng et al., 2022), climate (Taylor et al., 2012), elections (Straffin Jr, 1980; Le Pennec & Pons, 2023), molecular biology (Shaw et al., 2010), financial markets (Cont & Bouchaud, 2000), self-driving cars (Dosovitskiy et al., 2017), and robotics (Todorov et al., 2012). Simulation environments are invaluable in these applications as they enable the evaluation of theories or methods, on-the-fly decision-making for a system based on simulation, or the training of AI systems, for instance through reinforcement learning. As conversational interfaces see widespread adoption in society, we view user LMs as a foundational simulation environment for user-AI conversation.

**Simulating Human Users In Conversation.**  User simulation has received considerable attention in past research, serving as an environment for dialogue policies to interact with in reinforcement learning setups (Shi et al., 2019). Early conversation simulators were introduced in task-oriented dialogue research, consisting of rule-based systems that define a user goal and an agenda of pending actions that dictate the next user move (Schatzmann et al., 2007; Li et al., 2016). These systems later evolved to include neural-based components that encode the dialogue state and generate user actions (Asri et al., 2016; Kreyssig et al., 2018; Gür et al., 2018; Tseng et al., 2021). More recently, pre-trained assistant LMs have been adopted in user simulation research. The majority of existing works have prompted assistant LMs in a zero-shot manner to role-play a user (i.e., "*You are a user chatting with an AI assistant ...*"), instructing them to generate the next user turn given the dialogue history (Li et al., 2024; Luo et al., 2024; Zhong et al., 2025; Chang et al., 2025). However, a growing body of work highlights that these prompted assistants exhibit low alignment with real human interactions and often diverge from human behavior (Ivey et al., 2024; Yoon et al., 2024; Zhu et al., 2024).

A few studies have proposed fine-tuning approaches to generate user turns given the dialogue context (Liu et al., 2022; 2023; Sekulić et al., 2024) by training on task-oriented dialogue datasets such as MultiWOZ (Eric et al., 2019; Cheng et al., 2022). Such fine-tuning approaches have been mainly focused on generating synthetic user-assistant conversations that can be used as additional training data for assistant LMs (Wan et al., 2022; Ferreira et al., 2024; Dhole, 2024), with some focusing on generating diverse user profiles (Kong et al., 2024; Wang et al., 2025). Our work complements this body of literature by introducing general-purpose *user language models* trained to follow a defined user intent and simulate users in conversations with assistants. We show that our user LMs better align with real human behavior compared to prompting assistant LMs to role-play users.

**Evaluating User Simulators.**  There is little work on how to evaluate simulators at replicating human behavior in their conversations with assistants (Deriu et al., 2021). Some studies measure the correlation between simulator and human utterances using lexical or stylistic metrics (Ivey et al., 2024; Pietquin & Hastie, 2013), or the diversity of their generated utterances (Wang et al., 2025). Others have resorted to human annotation to judge the quality of the simulated conversations (Kranti et al., 2025; Sekulić et al., 2024; 2022; Shi et al., 2019). Another line of work evaluates the simulator more indirectly by testing whether task performance with the assistant correlates with performance in real user interactions (Chang et al., 2025). Our study introduces fine-grained evaluations of simulator properties related to multi-turn interaction behavior and robustness. We also assess simulators when deployed for solving coding and math tasks with assistants.

## 7  CONCLUSION

In this work, we introduced approaches to train and evaluate User LMs. Through many experiments, we showed the inefficacy of prompting assistant LMs to simulate users and highlighted the value of purpose-built user LMs, trained on user utterances from a large corpus of user-assistant interactions. We demonstrated a practical application of user LMs - using them to simulate conversations with an assistant. This surfaced limitations in current assistant LMs, which displayed degraded performance due to the more realistic simulation environment. We hope the release of UserLM-8b spurs further research by the community on user language modeling.

## ACKNOWLEDGMENTS

We would like to thank Chan Young Park, Jillian Fisher, and Serina Chang, and the AI Interaction and Learning team members at Microsoft Research for valuable discussions. We also thank Pangram for providing credits to access their API, facilitating some of our robustness evaluations. Tarek Naous and Wei Xu are partially supported by the NSF CAREER Award IIS-2144493. Any opinions, findings, and conclusions or recommendations expressed in this material are those of the authors and do not necessarily reflect the views of the National Science Foundation.

## ETHICS STATEMENT

The UserLM-8b we contribute in this work is trained from a base checkpoint (Llama3-8b-base), which has likely not undergone the conventional post-training mechanisms for output safety and alignment. Thus, our model is expected to behave similarly to a base model, which could be more prone to generating toxic language than an instruction-tuned model. We also note that our model is trained to generate user utterances that are realistic, which we also demonstrated to be not easily detectable by current state-of-the-art detectors of AI-generated text (§3). The purpose of our model is to be used for research purposes as a simulator for interactive evaluation of assistants and similar beneficial use cases. We hope our work encourages future research on building detectors that differentiate between utterances generated by a user LM and ones written by real humans.

## REPRODUCIBILITY STATEMENT

Our trained UserLM-8b model will be made publicly available for research purposes. The model was trained using publicly available data, enabling the community to reproduce and expand on our experiments. We report the decoding configuration we used for our UserLM-8b in Appendix C.1 and all the prompts we used in Appendix D, which are needed to replicate the results in our experiments.

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

# A  ADDITIONAL EXPERIMENTAL DETAILS

**WildChat Deduplication.**  During initial analysis, we noticed a lot of near-duplicate conversations in WildChat - often consisting of prompt templates with minor input variations. To improve the quality of the training corpus, we applied a 7-gram counter on the first-turn user prompts and manually examined the most frequent patterns. We identified several recurring prompt templates, with one used in as many as 81,196 conversations, that do not represent natural user interaction and could degrade model quality. Removing all near-duplicates reduced the dataset to 384,336 conversations. To verify the efficacy of our filtering and data splitting, we computed the 7-gram overlap between the first-turn user prompts in the train and test sets, finding only 4.19% overlap. This low overlap confirms that our deduplication and user-based splitting procedure effectively minimizes data leakage between splits.

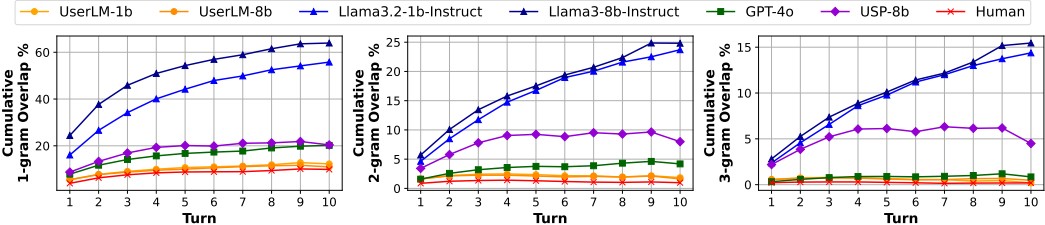

Figure 5: Cumulative n-gram overlap between generated user turns and the generic intent of each conversation. Results are averaged across each turn for all conversations in PRISM. Our user LMs achieve the lowest cumulative n-gram overlap with the intent, aligning with real human utterances.

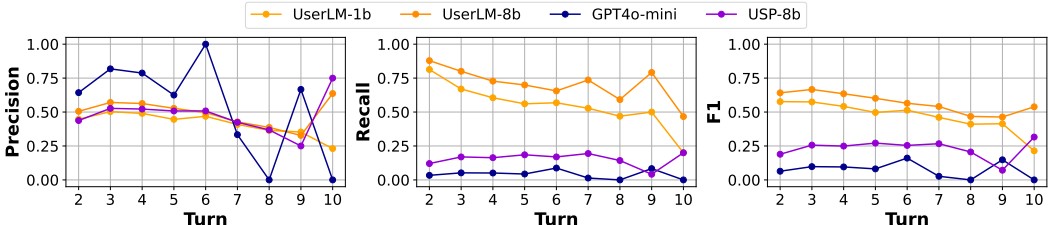

Figure 6: Precision, Recall, and F1 score per turn for our dialogue termination evaluation. Results are shown for models that achieved more than an average F1 score of 10.

## B ADDITIONAL RESULTS

### B.1 PER-TURN ANALYSES

**Perplexity.** Figure 4 shows the token-level PPL achieved by all models on the human utterances of the PRISM dataset. Our user LMs outperform the instruction-tuned models and the USP-8b baselines. Particularly, we find the largest PPL differences between our user models and all baselines to be at the first turn. This indicates that our user LMs are much better at initiating the first turn of a conversation the way real users would, but prompted instruction-tuned baselines struggle to do so.

**Intent Decomposition.** Figure 5 shows the cumulative 1-gram overlap across turns between the generated user turns and the generic intent we condition on. Our user models have the lowest overlap across turns, aligning closely with humans and outperforming the GPT-4o simulator that is explicitly instructed

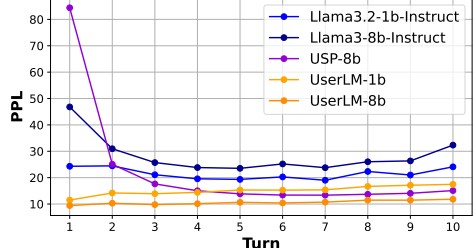

Figure 4: Per-turn token-level PPL achieved by models on PRISM utterances. All models are conditioned on the generic user intent of each conversation. Our user LMs outperform all baselines and achieve much lower PPL, especially at the first turn.

to split intent across turns and not to copy. The instruction-tuned Llama models show the highest overlap, starting at 25% which indicates higher levels of copying from the intent at the first turn.

**Dialogue Termination.** Figure 6 shows the precision, recall, and F1 score per-turn for the dialogue termination evaluation. We find that prompted assistants such as GPT4o-mini achieve high precision, yet very low recall, indicating that while their termination usually aligns with real users, they rarely do so and often proceed to continue the conversation. This behavior becomes worse with better assistants such as GPT-4o, which rarely terminates dialogues, achieving near-zero F1 score. Our UserLM-1b and UserLM-8b show a better balance of precision and recall, aligning better with human behavior. We note that in practice, decoding can be constrained in our user LMs to not generate the <|endconversation|> token if desired.

|  | | **Multi-Turn Interaction** | | | **Simulation Robustness** | | |
|---|---|---|---|---|---|---|---|
|  | Initialization | 💡(↑) | ⛓(↓) | ◈(↑) | ☕(↑) | ⚙(↑) | 🔒(↑) |
| UserLM-1b | Instruct | 89.02 | 8.40 | 45.21 | 53.79 | 91.13 | **98.00** |
|  | Base | **90.90** | **3.07** | **56.83** | **78.96** | **91.30** | 93.55 |
| UserLM-8b | Instruct | 89.04 | 8.88 | 56.96 | 56.54 | **95.45** | **97.75** |
|  | Base | **94.55** | **2.69** | **63.54** | **80.21** | 93.95 | 94.65 |
| Human | | 94.01 | 1.68 | — | 90.15 | — | — |

Table 4: Additional evaluations on intrinsic properties comparing our User LMs when training from a base checkpoints (Llama3.2-1b-Base and Llama3-8b-Base) vs instruction-tuned checkpoint (Llama3.2-1b-Instruct and Llama3-8b-Instruct). The user models trained from a base checkpoint achieve overall better results than ones trained from instruction-tuned checkpoints.

## B.2 STARTING CHECKPOINT ADDITIONAL ANALYSIS

Our earlier experiments showed how User LMs trained from a base checkpoint achieve better distributional alignment (lower PPL) on human utterances (§2.3), suggesting better simulators. We further verify this by evaluating models trained from either checkpoints on our intrinsic properties presented in §3. Table 4 shows the results, where we find that User LMs trained from the base checkpoints generally outperform ones trained from instruction-tuned models. Most notably, models trained from the base checkpoints are better at intent decomposition, dialogue termination, and produce text that more closely resembles the style of real humans. Models trained from instruction-tuned checkpoints show similar performance in terms of role and intent adherence. These results further confirm the findings from our initial PPL analyses, which show that models already trained to be assistants are more difficult to train to become user models.

## C  CONVERSATION SIMULATION EXPERIMENTAL DETAIL

### C.1  GENERATION CONFIGURATION FOR USERLM-8B

In the simulation experiment detailed in Section 4, we implemented a few simple generation guardrails to counteract the small nature of the model (8b parameters), as we found that simply sampling unfiltered responses from the model did not result in satisfactory quality necessary to conduct simulation experiments.

**Guardrail 1: Filtering First Tokens.**  In our experimental simulation setting, initial trials revealed that the UserLM-8b frequently generates responses that started with "I", "You", or "Here", which led to repetitive user utterances. We constrained the decoding by implementing a logit filter, in which a set of 6 tokens (3 words listed above in lower- and capitalized format) had their logits set to 0 for the first token, ensuring they are not sampled as first words. This simple modification was effective in practice at steering the model towards more useful generations.

**Guardrail 2: Avoiding Dialogue Termination.**  In our extrinsic evaluation, we prohibited the model from generating the `<|endconversation|>` token that we used in our earlier analysis on dialogue termination capability, by setting its probability of being generated to 0.

**Guardrail 3: Maximal and Minimal Length Threshold.**  The experiment we conduct specifically evaluates assistant behavior in multi-turn, under-specified conversations. To encourage such behavior from the user simulator, we filtered out any user utterance of more than 25 words, as we found through inspection that longer utterances tended to reveal the entire problem formulation, reducing the need for multi-turn interaction. In case the UserLM-8b generates a response longer than the set threshold, we discarded the response and generated a new one, until we obtained a response below the threshold. We also filter out responses that contain fewer than three words, though we note that such occurrences were rare in our inspection.

**Guardrail 4: Filter Verbatim Repetitions.**  Inspection of simulated conversations further revealed that the UserLM-8b can, on occasion, repeat a previous user utterance verbatim, or copy verbatim the original intent, which does not advance the conversation appropriately. We therefore filter out such responses and re-generate using the simulator until a novel response is generated.

The four guardrails detailed above were implemented to maintain a quality in the simulated conversations we include in our analysis. We believe that these guardrails are needed due to the small size of our model, leading to noisy responses. We hypothesize that the need for such guardrails would be alleviated with a larger and more performant user LM, which we hope will arise in the future as the community develops high-quality simulators.

### C.2  ADDITIONAL EXAMPLES OF SIMULATED CONVERSATIONS

The simulation experiments in Section 4 conducted ten simulations for each task instruction, allowing us to observe the diversity of simulations that can occur. For a given task, Figure 7 provides three simulated conversations using GPT-4o for the simulation, and three conversations simulated using UserLM-8b. The UserLM-8b-based conversations are more distinct from each other, with more unique terminology (e.g., "ignore" vs. "keep" or "translate" vs. "replace"), which we measure quantitatively by observing a higher pace and lexical diversity. This additional diversity in simulation leads to a more challenging environment for the assistant, which is measured to have lower performance than when conversations are simulated using GPT-4o.

### C.3  DEFINITION OF SIMULATION EVALUATION METRICS

We now define the eight metrics used as part of our analysis in Section 4 that involves the simulation of multi-turn conversations using various user simulators.

Each simulated conversation in the experiment is based on an instruction that the user provides to the assistant from the Lost in Conversation corpus (Laban et al., 2025), which we use as the user intent of the simulation (i.e., "You are a user chatting with an assistant language model to complete

## SIMULATING USERS IN CONVERSATIONS

**... USING AN ASSISTANT LANGUAGE MODEL**          **... USING A USER LANGUAGE MODEL**

**User Intent:** Write a Python function: given an array of integers, sort ones between 1 and 9 inclusive, reverse the array, and replace digits by their name from "One", "Two", "Three", etc.

**GPT-4o**

| | | |
|---|---|---|
| Turn digits into names in a list | Start with a list of numbers. | Start by having a list of numbers. |
| I want to sort the numbers if they're between 1 and 9 | Only want numbers from 1 to 9, sort them if needed. | Next step is to sort numbers if they're between 1 and 9. |
| I want to flip the list after that | Now I want to flip the list around after sorting it. | Next, we can flip the list around after sorting it. |
| I want to use names for numbers from One to Nine | I want it to use the words for numbers, like One to Nine. | Also, I want to use words for the numbers from One to Nine. |

**UserLM-8b**

| | | |
|---|---|---|
| write a python function that accepts an array of integers | Sort the list of ints between 1 and 9 using the merge sort algorithm. | ignore any number that is not in the 1-to-9 range |
| keep only the integers between 1 and 9 | | now add sorting into it |
| sort those kept integers | Reverse the list. | now reverse the sorted list |
| Reverse that sorted list | Replace each digit by its corresponding name from "One", "Two", "Three", "Four", "Five", "Six", "Seven", "Eight", "Nine". | translate each remaining digit into its capitalized English name |
| translate each remaining digit into its English name, from 'One' through 'Nine' | | |

Figure 7: Selected simulated conversations for a fixed instruction, using a prompted GPT-4o (three simulations on the left) and using UserLM-8b (three simulations on the right). We only show user turns, and omit assistant responses. UserLM-based simulations are more diverse, using more varied language in each, as confirmed quantitatively bu our metrics-based analysis.

the following: [INSTRUCTION]"). Each instruction is declined as a *fully-specified instruction*: a single utterance that defines the instruction in its entirety, and a *sharded instruction*: a set of 3-8 "shards" that taken together, reveal the same information as the fully-specified instruction.

For our analysis, we consider shards as atomic units of information from the original fully-specified instruction, which enables us to scrutinize which information is and isn't revealed by a user simulator. Given a simulated conversation based on an instruction, we prompt OpenAI's o3 model to assign which specific shard (or shards) are revealed in each user turn. Manual inspection of 5 simulated conversations revealed that o3's mapping ability was accurate to conduct our analysis. The `Intent Coverage` metric is then defined as:

$$\text{Intent Coverage} = \frac{len(set(\text{revealed\_shards}))}{\#\text{ shards}}, \tag{1}$$

in other words, intent coverage measures the fraction of shards that were revealed at least once during simulation.

Sharded instructions have additional metadata attached to each shard, namely whether the shard is *required* or not to solve the task. This Boolean tag enables us to analyze which parts of an instruction they focus on. First, we calculate:

$$\texttt{Repeat Required} = \begin{cases} 0 & \text{if all required shards are revealed at most once,} \\ 1 & \text{otherwise (i.e., at least one required shard is repeated 2+ times)} \end{cases} \tag{2}$$

In other words, this measures the percentage of conversations in which the user simulator highlighted important information by repeating it across turns.

We also compute:

$$\texttt{Skip Non-Required} = \begin{cases} 0 & \text{if all the non-required shards are revealed,} \\ 1 & \text{otherwise (i.e., at least one non-required shard is ommitted)} \end{cases} \tag{3}$$

which measures whether the user simulator is selectively picking secondary information to skip over during simulation. Non-required shards are typically obvious information (such as a trivial base case or a clarifying example).

Finally, we used o3 to identify any information introduced by the simulator in the user utterances that was not initially present in any of the shards. This information is then used to compute:

$$\texttt{Additional Demands} = \begin{cases} 0 & \text{if no novel information is in the simulated conversation,} \\ 1 & \text{otherwise (i.e., 1+ user utterance introduces novel information)} \end{cases} \tag{4}$$

In Section 4, we perform qualitative analysis of the additional demands introduced by user simulators, which are categorized into three main types: providing examples, defining naming conventions for functions or variables, and adding implementation or algorithmic constraints.

The above-defined metrics are all computed for each individual simulated conversation and then averaged across each user simulator. In Table 3, each result therefore represents the aggregation of 650 simulations (65 instructions, and 10 simulations per instruction).

The next metrics we define require multiple simulations for a given instruction: given that the simulators are probabilistic in nature (i.e., language models run at temperature $T = 1.0$), each simulation diverges and is unique. We simulate a total of 10 conversations ($C_1, C_2, \ldots C_{10}$) for each instruction for each simulator, and compute the following metrics based on the set of simulations obtained for each instruction:

$$\text{Turn Variance} = var_{i=1}^{10}(\#turns(C_i)), \tag{5}$$

where $var(X)$ is the empirical estimate of variance of the series, and #turns is the number of conversational turns (user-assistant exchanges) in the simulated conversation. We also report the range of the number of turns, to help elucidate whether an increase in variance comes from the lower or upper range of conversation turns. The range is computed as:

$$\text{Turn Range} = [min(\#turns(C_i)), max(\#turns(C_i))], \tag{6}$$

for each instruction. We compute lexical diversity following prior work (Stasaski et al., 2020; Tevet & Berant, 2020) by calculating pairwise-lexical overlap of unigrams:

$$\text{Unigram Overlap} = \frac{1}{10C2}(\sum_{i \neq j} \frac{|intersection(C_i, C_j)|}{|union(C_i, C_j)|}), \tag{7}$$

In more specific detail, for each pair of simulated conversations on a common instruction, we extract the lemmatized unigrams of all user utterances and measure the Jaccard index. We average this across all pairs of simulated conversations, effectively measuring the average overlap in wording across two simulated conversations based on the same instruction. We then calculate the complement:

$$\text{Unigram Difference} = 1 - \text{Unigram Overlap}, \tag{8}$$

which assesses the simulator's ability to use unique wording in each conducted simulation.

Turn variance, turn range, and lexical diversity are computed for each instruction (based on 10 simulations), and aggregated across instructions.

As a final metric, we assess whether the assistant successfully completes the task (i.e., generates Python code that solves held-out unit tests or finds the correct mathematical solution to the problem), which is computed as:

$$\text{Assistant Score} = \begin{cases} 1 & \text{if an assistant solution at any turn passes evaluation,} \\ 0 & \text{otherwise (i.e., none of the attempted solutions are correct).} \end{cases} \tag{9}$$

This final metric does not directly evaluate the user simulator, but instead measures the effect of the simulator on the downstream performance of the assistant (GPT-4o), giving us insight into the impact that the user simulator has on the task-solving ability of the assistant.

| Metric | Task: Math | | | Task: Code | | |
|---|---|---|---|---|---|---|
| | 4o-mini | GPT-4o | UserLM-8b | 4o-mini | GPT-4o | UserLM-8b |
| | **Intent Coverage** | | | | | |
| Intent Coverage (%) | 95.1 | 97.1 | 76.5 | 82.8 | 79.1 | 76.6 |
| | **Information Diversity** | | | | | |
| %Repeat Required | 14.5 | 44.0 | 3.0 | 39.6 | 12.2 | 58.9 |
| %Skip Non-Required | – | – | – | 10.9 | 14.6 | 37.7 |
| %Add Demands | 10.0 | 0.5 | 40.2 | 9.3 | 1.1 | 45.5 |
| | **Pace Diversity** | | | | | |
| Turn Variance | 0.4 | 0.1 | 3.4 | 1.2 | 0.9 | 2.6 |
| Turn Range | 4.0-5.3 | 5.3-5.4 | 2.0-6.8 | 3.5-5.9 | 3.5-5.4 | 2.2-6.7 |
| | **Lexical Diversity** | | | | | |
| Unigram Difference | 0.32 | 0.30 | 0.68 | 0.47 | 0.43 | 0.72 |
| | **Assistant Performance** | | | | | |
| Assistant Score | 0.83 | 0.85 | 0.46 | 0.69 | 0.70 | 0.63 |

Table 5: Summary of evaluation metrics for simulated conversations across two tasks types (**math** and **code**) with three user simulators (prompted assistant LMs and trained user LMs). Each simulator is evaluated on intent coverage, information diversity, conversational pace, lexical diversity, and downstream assistant performance.

## C.4   SIMULATION RESULTS PER-TASK

Our main results presented in Section 3 aggregate metric scores across the coding and math solving tasks to give a sense of the simulator behavior in general. We show the results for each of those tasks in Table 5. We note that for the math instructions, all shards were required, rendering the calculation of `Skip Non-Required` not possible.

Overall, we observe similar trends across the two tasks. Some noteworthy differences: on math tasks, the GPT-based user simulators stick to the script more strictly, with almost 100% coverage of underlying shards, almost no turn variance, and a higher gap in observed assistant performance with the UserLM-8b. On the other hand, all simulators exhibit a little more diversity in their responses in the coding tasks, with better lexical and turn diversity.

# D  PROMPTS

**Intent Generation for Conversations.**   Figure 8 shows the prompt we used to generate a generic user intent for each conversation in Wildchat and PRISM datasets. We provide GPT-4o with the full conversation between the user and the assistant and instruct it to create a short summary of the high-level intent of the user from the conversation, without mentioning any specific details. We provide 3-shot examples where we manually wrote the generic user intent from the conversation.

**User Simulation with Assistant LMs.**   Figure 9 shows the prompt we used to simulate users with assistant LMs, which we based on how recent studies have done it (Chang et al., 2025; Ivey et al., 2024). The assistant is given the generic intent of the user and instructed to generate the user utterance at the first turn and subsequent turns given the conversation history so far. We further engineered the prompt to provide additional instructions on the desired behavior of user utterances (i.e., intent decomposition, making occasional typos, not writing too verbose utterances, etc.), without overdoing these phenomena.

**LM-as-Judge for Intent Adherence.**   Figure 10 shows the prompt we used to classify if the user simulator successfully adhered to its intent in our intent adherence evaluation. We provide GPT-4o with the user simulator question to the assistant, the assistant's refusal to provide the answer and suggestion of looking at something else, and the user simulator's reply. Based on this conversation history, we instruct GPT-4o to classify if the user simulator stuck to its intent by reiterating what they asked in their first turn, or whether they complied with he suggestion of the assistant.

---

**Prompt Template for Intent Generation**

```
You are given the conversation history between a user and assistant model and your task is to create
a summary of the user's intent from the conversation.

Your summary should be structured to define what the high level intent of the user is, but
should not go into specific details.

Format the summary to start with "You are a user chatting with an assistant language model to"

Example 1:

Conversation History:

<user>: List to me 5 beautiful sights in Algeria

<assistant>: Algeria is a country of striking natural beauty and rich cultural heritage.
Here are five breathtaking sights you should know about:
Tassili n'Ajjer National Park, Djemila, Timgad, Beni Hammad Fort, Basilique Notre Dame d'Afrique

Intent Summary:

You are a user chatting with an assistant language model to obtain some recommendations of places
to see in Algeria.

Example 2:

Conversation History:

<user>: does allertine cause weight gain

<assistant>: As an AI language model, I don't have personal experiences. But according to the
available literature, Allertine is not known to cause weight gain. However, ...

<user>: do antihistamines cause weight gain

<assistant>: Most antihistamines do not typically cause weight gain as a direct side effect. However,
some  people find that they experience increased appetite or cravings for sugary, high-calorie ...

<user>: how much weight can i lose by cutting sugar for 40 days

<assistant>: The amount of weight you can lose by cutting sugar from your diet for 40 days will depend
on a variety of factors, including your starting weight, age, gender, level of physical activity, ...

Intent Summary:

You are a user chatting with an assistant language model to get information about strategies for
losing weight and the impact of certain drugs or medications on weight gain.

Example 3:

Conversation History:

<user>: Is einstein determinism challenged by quantum mechanics?

<assistant>: Quantum mechanics challenges Einstein's determinism, which held that the universe is fully
predictable given complete information. Key quantum principles like the Heisenberg uncertainty  ...

<user>: What did hawkins say?

<assistant>: It seems you are referring to Stephen Hawking, the renowned physicist. Stephen Hawking
supported the principles of quantum mechanics, believing that the universe is fundamentally ...

<user>: So does it mean determinism is refuted?

<assistant>: Determinism is not entirely refuted but is significantly challenged at the quantum level,
where subatomic behavior follows probabilistic principles rather than predictable, classical laws ...

<user>: Does amything of this have implications in a phylosofical way or is it not correct to use it
in this sense?

<assistant>: Quantum mechanics challenges classical determinism and raises important philosophical
questions in metaphysics, free will, and the nature of observation. Its probabilistic nature  ...

Intent Summary:

You are a user chatting with an assistant language model to understand how quantum mechanics
challenges Eistein's determinism and get the perspective of different scientists on this.

Now generate a summary of the user intent for the following conversation:

[CONVERSATION]

Reply with only the intent summary and nothing else.
```

Figure 8: Prompt template used to generate generic user intents from conversations with GPT-4o. We give GPT-4o three hand-engineered examples (assistant turns shortened for brevity).

---

**Prompt Template for User Simulation with Assistant LMs (First Turn)**

```
You are a human user interacting with an AI system to [INTENT].
Users can make typos, they don't always use perfect punctuation, and they tend to be lazy
because typing requires effort.
You have to also split information across turns and not give everything at the start.
However, you should not make overdo these things in your outputs, you must realistically act
like a human.

Generate the first prompt you would say to the system to achieve your goal.
```

---

**Prompt Template for User Simulation with Assistant LMs (Subsequent Turns)**

```
You are a human user interacting with an AI system to [INTENT].
Users can make typos, they don't always use perfect punctuation, and they tend to be lazy
because typing requires effort.
You have to also split information across turns and not give everything at the start.
However, you should not make overdo these things in your outputs, you must realistically act
like a human.

Here is your conversation so far with the AI system:

====================================================

[CONVERSATION HISTORY]

====================================================

If your goal of your conversation is answered by this response, respond ONLY with "<|endconversation|>"

If not, generate the next prompt you would say to the system to achieve your goal.
```

Figure 9: Role-playing prompt template used to instruct assistant LMs to generate the next user turn.

---

**LM-as-Judge Prompt Template for Intent Adherence**

```
You are given a conversation between a user and an AI system.

User Question: {question}
AI Response: {refusal}
User Reply: {output}

Classify if the user accepted the AI's suggestion or repeated their original question.

If the user's reply repeats or rephrases their original question, output only: REFUSED
If the user's reply follows the AI's suggestion, output only: ACCEPTED
```

Figure 10: Prompt used to judge if a simulator sticks to its intent in our intent adherence evaluation.

