# OpenReview forum: "Flipping the Dialogue: Training and Evaluating User Language Models"
_ICLR.cc/2026/Conference — ICLR 2026 Poster_

### Official Review · Reviewer_YAMu · 2025-10-29

**Soundness:** 1
**Presentation:** 2
**Contribution:** 2
**Rating:** 2
**Confidence:** 3

**Summary:**

This paper proposes UserLM to evaluate LM performance in realistic multi-turn conversation settings. The authors train UserLMs on flipped dialogues between users and LMs to simulate users, given user intents. The experiments show that UserLMs' simulations better match actual human responses and remain within the user role. As a result, the performance of assistant LMs is lower against UserLMs than against prompt-based baselines.

**Strengths:**

1. This paper addresses an important problem of improving and making more realistic user simulations for chat-based LM evaluation.

2. The proposed approach, termed UserLM, provides better user simulation that matches human responses in various aspects, whereas previous prompt-based approaches fail.

**Weaknesses:**

1. The paper does not provide evidence that better user simulation leads to better assistant LM evaluation. It seems the assistant LM performs worse against UserLM. However, it is not clear whether this is because real user requests are more challenging or because UserLM has some unexpected traits that confuse the assistant. Moreover, this does not mean that assistant LM evaluation based on UserLM offers better representation of the assistant's quality.

2. The WildChat dataset seems to contain many almost-duplicates that are not necessarily from the same IP addresses. This may be due to popular prompts shared by many users or to a single user using multiple IP addresses. Without careful data splitting to account for this, there may be data leakage into the test sets.

**Questions:**

1. What are the findings that are not available in other user simulation approaches but that UserLM enables?

2. Does an assistant LM performing worse against UserLM suffice to conclude that UserLM-based simulation offers better evaluation? How can we rule out that UserLM is just confusing the assistant LM?

3. Were there any additional checks of data leakage into the test sets?

---

> ### Author Response · Authors · 2025-11-17
> **Response to Reviewer YAMu 1/2**
>
> Thank you so much for your efforts reviewing our paper, we really appreciate it! We respond to your concerns and questions below.
>
> **(Q1) WildChat Deduplication:** We clarify that near-duplicate non-natural exchanges are indeed a concern in WildChat as we discuss in Section 2.2 and Appendix A. To address this, we applied a n-gram based filtering method, deduplicating any conversations with frequent 7-grams. We note that this was done across the whole dataset before splitting users based on their IP addresses. This led to removing roughly 20% of WildChat conversations, favoring quality over volume. We have run a secondary analysis to evaluate the level of duplication in the dataset, finding only 4.19% of overlap in 7-grams between the train and test sets, confirming the efficacy of our filtering. We believe that this additional analysis confirms the validity of our data split, which was carefully derived, and will add this additional analysis to Appendix A.
>
> We also note that our evaluations are not limited to Wildchat test set, but we also use PRISM which is a smaller-scale dataset collected through a different protocol (active recruitment) and with a different (less noisy) population of users. In Table 1, we find that we significantly improve distributional alignment with these out-of-domain users (since we did not train on any PRISM data), indicating generalization and lowering the chance that gains in perplexity are due to overfitting on noisy patterns specific to WildChat. The PRISM experiment was precisely intended to assess generalization to user populations, as you point out.
>
>
> **(Q2) Drop in assistant performance:** We agree that there could be multiple causes for degradation in assistant performance. However, we clarify that we do not look at performance degradation by itself to make our conclusion, but in conjunction with the fine-grained analyses that we performed to verify that this is indeed due to the more realistic and challenging simulation environment that UserLM provides. The evidence from this is due to two main things:
>
>
> 1) The level of intent coverage of UserLM (first row of Table 3) is similar to the prompted assistants: so the UserLM has revealed information necessary to solve the task similarly to prompted assistants (therefore the assistant is not failing simply because the UserLM fails at revealing task information).
>
> 2) The level of information and lexical diversity of user turns generated by UserLM (displayed in Table 3) is higher than the simulators based on prompted assistants, indicating more varied vocabulary that aligns with how real users chat with assistants (which we analyzed in our earlier intrinsic evaluations).
>
> - We also note that we provide three example conversations using UserLM on the same task intent in Figure 7 of the Appendix that illustrate this, where each user conversation exhibits more lexical diversity with different conversations using varied vocabulary (keep vs. ignore vs. filter) and pace, which is what leads to an increased challenge for the assistant counterpart. We realize this is an important implication of our findings (the cause of assistant degradation), and we will add a subsection to our Discussion section (Appendix B), as currently this point must be inferred from findings that are scattered across the paper. Thanks for pointing this out, we would be happy to know if you had any other suggestions!

---

> > ### Author Response · Authors · 2025-11-17
> > **Response to Reviewer YAMu 2/2**
> >
> > **(Q3) Findings not available in other user simulation approaches:** We clarify that our UserLMs provides a more realistic simulation environment compared to two main types of prior work on user simulation:
> >
> > - The first (and larger) one is prior work leveraging prompted assistant models to simulate users. Our evaluations indicate that such methods are not satisfactory, with the simulators lacking ability to align with the distribution of utterances (Table 1) and not exhibiting important user-like properties (Table 2). We build a smaller specialized model we show outperforms the prompting of larger assistant models. Most importantly, we find that stronger assistants are not necessarily better user simulators, indicating that improved assistants will not lead to strong user simulators. For example, our UserLM-8b was better at simulating realistic user conversations than prompting GPT-4o to simulate a user.
> >
> > - The second body of work has trained user simulators primarily for the creation of synthetic corpora used to train assistant models (such as the USP-8b model we compare to). In other words, the user simulators they trained were more of a byproduct for another objective (generating synthetic conversations to train assistants), whereas we focus on building a simulator that is steerable towards a particular task. This required us to set up intrinsic evaluation (Section 3) tying desirable properties to quantitative metrics, allowing us to evaluate progress on the task of user simulation itself. In the discussion section, we discuss the potential uses of user simulators beyond evaluation of assistant models, they include: user modeling (in statistics & survey studies), LLM-as-a-judge, and synthetic data generation.
> >
> > We hope our response addressed your concerns and questions and we would really appreciate it if you could increase your score!

---

> > > ### Author Response · Authors · 2025-11-25
> > >
> > > Dear Reviewer YAMu,
> > >
> > > We wanted to send you a gentle reminder to check our rebuttal comments. We entirely understand that you are likely busy, and appreciate your time. As our most critical reviewer, we prioritized responding to your raised concerns, to allow for a discussion.
> > >
> > > Thank you again for your time.

---

> > > > ### Comment · Reviewer_YAMu · 2025-11-25
> > > >
> > > > Thank you for your detailed answers to my review. The rebuttal addresses my concern on deduplication, but I still have concerns on validity (reason for the drop in assistance performance) and significance (findings enabled by UserLM) of this work.
> > > >
> > > > I still think there should be a more comprehensive evaluation of UserLM that covers additional aspects beyond intent coverage and lexical diversity. Is UserLM fluent in language usage? How about the difficulty of the tasks requested?
> > > >
> > > > Moreover, even if UserLM provides a more realistic simulation of users, it is unclear why such a simulation is necessary for LLM quality estimation. What new observation was enabled by UserLM, and how does it follow from the realistic-ness of UserLM? If LLM A performed better than LLM B in the UserLM-based evaluation, wouldn't it also be true in the existing, less-realistic-simulation-based evaluation?

---

> > > > > ### Author Response · Authors · 2025-11-27
> > > > >
> > > > > Thanks so much for responding! We provide further clarification below:
> > > > >
> > > > > **Evaluation:** We would like to clarify that we performed many types of evaluations in the paper which go beyond intent coverage and lexical diversity. In section 3, we proposed several additional evaluations for properties that are desirable for a user simulator. For example, we proposed three types of evaluations that focus on the robustness of the simulator that answer the following questions:
> > > > >
> > > > >
> > > > > *How closely does the language of the simulator resemble that of real humans?* We show how state of the art detectors of AI generated text (Pangram) fail to detect user utterances generated by UserLM, misclassifying them as being human-written. On the other hand, user utterances generated by GPT-4o when engineered to roleplay a user are still easily detectable as being generated by an AI model. This shows that UserLM can generate fluent text that closely resembles the style of real human users, which further validates our perplexity analyses of Section 2.3.
> > > > >
> > > > >
> > > > > *How well does the simulator stick to its role as the user?* It is known that prompted assistants as user simulators often don’t stick to their user role in a multi-turn conversation, which is due to their inherent assistant behavior that they are post-trained to exhibit. This is a critical aspect of robustness for the simulator, which we proposed an evaluation setup for, showing how UserLM is much better at role adherence, whereas a GPT-4o-based simulator fails to do this in 60% of cases.
> > > > >
> > > > >
> > > > > *How well does the simulator stick to its task intent it wants to solve?* Another important aspect of robustness is that the simulator must stick to what it intends to solve, and not adhere to irrelevant suggestions of the assistant when they are confused. Prompted assistants often fail at this, due to their agreeable nature, with the GPT-4o-based simulator failing in 30% of cases. On the other hand, UserLMs are much better at this.
> > > > >
> > > > > Besides such intrinsic evaluations of different important properties in Section 3, we have performed extrinsic evaluations in Section 4 to further evaluate and understand the value of UserLM when deployed to interact with an assistant for solving coding and math tasks in multi-turn conversations.
> > > > >
> > > > >
> > > > > **Why realistic simulators are needed for LLM quality estimation:** There are multiple past papers that we have cited in our introduction and related work that demonstrate how the performance of LLMs drops when solving tasks during multi-turn conversations with real users. Typically, LLMs are evaluated on single-turn benchmarks, where the full task information is given in a single prompt. However, this is much different than the setup in which assistant LLMs are commonly used in the real world, where they are interacting with users in multi-turn conversations to solve tasks, users rarely provide the full task information or intent in a single prompt, but rather do it in multiple turns.  Past papers have had to hire real human users to interact with LLMs in order to show how their quality deteriorates in multi-turn setups. However, it is not scalable nor cost efficient to always hire human users, which is why realistic and robust user simulators are needed to make progress in the field and improve the abilities of LLMs in multi-turn setups. This is what our paper aims to achieve: we produce more realistic user simulators and propose ways to evaluate their desired properties.
> > > > >
> > > > > Our extrinsic evaluations in Section 4 show that when you simulate valid conversations using UserLMs vs baseline simulators (prompted GPT-4o and GPT-4o-mini), the assistant fails to solve the tasks more often, despite all the task information being given by all simulators. The purpose of the intent coverage in Table 3 is to validate that the task information is being provided by the simulator (which all simulators perform well on). However, it can be seen from the other metrics regarding the type of information provided and the pace of the conversations that UserLM is introducing nuances a real user would do in the real world (skipping obvious details, providing additional constraints, etc.). On the other hand, the baseline simulators lack this behavior and are monotonous: they always simulate very similar conversations with little novel nuances, which is less useful for estimating how the assistant would perform in the real world. This contrast hopefully shows why a more realistic simulator is necessary - it will better reflect how things will go in the real world, and give us a better understanding on how good LLMs will be at handling these scenarios.
> > > > >
> > > > > We hope that this addresses your concerns and provides further clarity regarding the comprehensiveness of our evaluations and why a more realistic simulator is needed to better estimate the quality of LLMs in real world multi-turn setups. Please let us know if certain things are still unclear!

---

### Official Review · Reviewer_ep5V · 2025-10-31

**Soundness:** 2
**Presentation:** 3
**Contribution:** 2
**Rating:** 2
**Confidence:** 4

**Summary:**

This paper argues that assistant LMs, which are post-trained to produce exhaustive, unambiguous replies, do not faithfully mimic messy, indirect human user behavior in multi-turn dialogues. It shows that using assistant LMs as user simulators is flawed; stronger assistants make worse simulators. This paper propose “User LMs,” post-trained specifically to simulate human users, and report that these models align better with real user behavior and yield more robust simulations.

**Strengths:**

- This casts doubt on current practice and suggests the need for a user simulator better aligned with real users.
- Extensive experiments show that the proposed post-trained user simulator better captures the properties targeted by the designed metrics.

**Weaknesses:**

Evaluation Metrics
* While mimicking user behavior can be useful for building a user simulator, it is not clear that the properties highlighted as important for user behavior are actually critical for evaluating an LM assistant.

 * The results in Table 1 may offer limited insight: a model trained on conversational data should have lower perplexity on test sets from the same distribution (WildChat) or similar data (PRISM).

 * An intent (e.g., ‘You are a user chatting with an assistant language model … medications on weight gain.’) provides strong topical context. Given such a prompt, the conversation will predictably be steered toward that topic, which in turn reduces perplexity.

* Some evaluation metrics need stronger justification, as they do not appear to faithfully capture what is required for realistic user simulation.
    * There are many potential, confounding reasons for lower assistant performance. Is it reasonable, therefore, to conclude that lower performance indicates the user simulator is better aligned with real users?
    * Why is the simulator’s ability to end the conversation considered so important for evaluating multi-turn QA with LLMs?
    * Since the user’s response depends on the assistant’s response, is it appropriate to compute turn variance without accounting for this dependency?
    * The claim in Intent Decomposition section that “a lower overlap is particularly desirable because it indicates that the model expresses its intent using varied language while introducing details progressively” is not substantiated.

    * The experimental setup feels somewhat artificial. The methods for evaluating User-role adherence and Intent adherence seem overly narrow, specific, and somewhat unlikely to occur in practice.

Real-User Alignment
* What exactly constitutes a “real user” in this work? Was any human study conducted to validate that definition?
* As users become more accustomed to LLM chatbots and adapt their behavior to use them effectively, is a new human study needed to compare the proposed simulators with up-to-date real user behavior?
* What is the practical value of building the user simulator? Do you have empirical evidence showing that it improves the evaluation of LLM performance or leads to better model outcomes?

**Questions:**

Please see the Weaknesses section.

---

> ### Author Response · Authors · 2025-11-17
> **Response to Reviewer ep5V 1/3**
>
> Thank you so much for your efforts reviewing our paper, we really appreciate it! We respond to your concerns and questions below.
>
> **(Q1) Perplexity generalization on the test set:** We agree that measuring generalization only on the test set of WildChat (a held-out set of users) is limiting, as perhaps it represents a narrow range of users that limits the analysis. We clarify that this is the reason that we also include the PRISM dataset in our experiments which is quite different from WildChat: it was collected by a different research group, at a different time, with paid participants (on Prolific vs. unpaid volunteers in WildChat), and using a different study protocol. We therefore believe that observing lower perplexity (half of the best other baseline) on this out-of-distribution test set is a promising indication of generalization. We note that the fact that PRISM perplexities are generally higher than WildChat test are indicative of dataset differences. We will clarify this in our draft.
>
> **(Q2) Task intents reduce perplexity:** This is indeed a correct observation, and a desirable property. We clarify that the task intents we generate on are paraphrased, high-level summaries of the user goal from the conversation, which do not contain specific details from the conversation that could lead the model to simply copy content from the intent and naturally reduce PPL. The purpose of the intent is to be able to steer the model more efficiently towards a particular task while generating text that aligns with the distribution of real users. In Table 1, we report performance both without and with task conditioning, and indeed the observation is that   UserLMs achieve better alignment across both setups. In short, the UserLM is better at leveraging the signal in task intents compared to other models, and is therefore more likely to produce realistic conversation simulation for a desired task (it is more steerable). We will make this more clear in the paper to avoid confusion, thanks for pointing it out!

---

> > ### Author Response · Authors · 2025-11-17
> > **Response to Reviewer ep5V 2/3**
> >
> > **(Q3) Intrisic evaluation metrics:** We wholeheartedly agree that setting up quantitative metrics to evaluate user simulators is a challenging (and new) task. We believe we are the first ongoing work to propose 6 metrics specific to user simulation (for the setting of human-AI chats). We hope the community can take the metrics we propose and improve/extend on them, like everything in research the first attempt is never perfect. More specifically:
> >
> > - *Why end of conversation matters:* Users rarely provide explicit feedback on their satisfaction, and a user exiting a conversation is typically indicative of either a completed intent or dissatisfaction. Therefore being able to predict whether a user will end the conversation is a useful feedback signal. Furthermore, one could imagine that when deploying an assistant interface (like ChatGPT), one could generate multiple alternative responses, and serve the one for which a user simulator is most likely to predict a conversation continuation (if looking to maximize user engagement and conversation continuation).
> >
> > - *Turn variance:* We clarify that since the assistant always answers a user's utterance, the number of turns in the conversation is directly determined by the user simulator (depending on how much content from the intent it reveals and its willingness to continue the conversation). We also would like to point out that in Section 4 we fixed the assistant to GPT-4o for all experiments, so that the choice of assistant does not impact metrics (as you point out).
> >
> > - *Intent Decomposition:* we agree that our definition of intent decomposition is a little confusing. We propose to more clearly describe it with the following: “The current task intents use generic language that does not match the language of real user utterances, as can be seen from very low overlap n-gram overlap between test intents and real user utterances (last column of Table 2).” We clarify that we find UserLMs to be better at rephrasing generic task intents, whereas prompted assistant LMs are more likely to use verbatim language from the task intent, over-indexing on provided language and not adequately adapting it to become a realistic user utterance. We hope this improved definition is clearer.
> >
> > - *Robustness tests:* We clarify that the aim of our robustness test is to evaluate whether the simulator can adhere to the user role and to its task intent, which are important properties that simulators tend to sometimes fail at in real deployment. We agree that our evaluation setup are simplified probes, and not intended as realistic assessments that might occur during large-scale simulation. We however argue that if non-robust behavior is observed even in these simplified settings, it is likely to occur in real, more complex settings as well. For example, when GPT-4o is prompted to simulate a user, but is then nudged (during conversation) to take on the role of the assistant, it does so more than 60% of the time. This indicates that the underlying model is only shallowly behaving like a user, and will frequently behave as an assistant when it believes it is required. In short, the probes are simple, and likely not sufficient to prove robustness, but likely necessary and we hope they can inspire more complex and realistic setups in the future.
> >
> > - *Drop in assistant performance:* We agree that there could be multiple causes for degradation in assistant performance. However, we clarify that we do not look at performance degradation by itself to make our conclusion, but in conjunction with the fine-grained analyses that we performed to verify that this is indeed due to the more realistic and challenging simulation environment that UserLM provides. The evidence from this is due to two main things:
> >
> > 1) The level of intent coverage of UserLM (first row of Table 3) is similar to the prompted assistants: so the UserLM has revealed information necessary to solve the task similarly to prompted assistants (therefore the assistant is not failing simply because the UserLM fails at revealing task information).
> >
> > 2) The level of information and lexical diversity of user turns generated by UserLM (displayed in Table 3) is higher than the simulators based on prompted assistants, indicating more varied vocabulary that aligns with how real users chat with assistants (which we analyzed in our earlier intrinsic evaluations).
> >
> > - We also provide three example conversations using UserLM on the same task intent in Figure 7 of the Appendix that illustrate this, where each user conversation exhibits more lexical diversity with different conversations using varied vocabulary (keep vs. ignore vs. filter) and pace, which is what leads to an increased challenge for the assistant counterpart. We realize this is an important implication of our findings (the cause of assistant degradation), and we will add a subsection about this to our Discussion section  (Appendix B).

---

> > > ### Author Response · Authors · 2025-11-17
> > > **Response to Reviewer ep5V 3/3**
> > >
> > > **(Q4) Definition of a real user and human study:** We clarify that in our paper, we define the realisticness of a user simulator by how much they align with the distribution and fine-grained properties of text written by human users in real user-LLM conversations such as WildChat. We believe that conducting a user study to evaluate user simulators is challenging. For example, presenting conversations to a reader and asking them to judge the realisticness is challenging, as judging human-ness is likely subjective. We however believe that we can collect user utterances for *held out users* that have had conversations with an assistant, and observe to what extent our user simulator is distributionally aligned with such a user. This is in fact what we do in our experiment with the held-out PRISM dataset: we observe that we are more able to predict the utterances of these held out participants than other methods, and therefore this participant population validates our user simulator. We therefore view that one potential way to continue validating user simulators is to hire participants to have conversations with chatbots, and study the distributional alignment between the user simulator and these new participants. We would like to add this framing to our discussion section. We are open to other ideas regarding human studies that could validate the quality of user simulators, as this is a novel research domain.
> > >
> > > **(Q5) Regarding distribution drift over time as users adapt the way they talk with AI:** This is an excellent observation, which can be studied with UserLMs. For instance, WildChat has recently released a larger version of its dataset with conversations that have occurred over a larger time window. One could train on sub-windows, and observe how a UserLM trained on 2024 conversations performs on 2025 conversations and vice-versa. This would allow us to quantify levels of drift and adaptation that have occurred, and also provide insights into whether online learning is required for up to date user simulation. We would like to add this to our discussion section as well, as it relates to other areas of ML (for instance on drift for NER over time, etc.), thanks for pointing it out!
> > >
> > > **(Q6) On the practical value of building user simulators:**  We believe interactive evaluation of assistants will be one key direction for practical value, which we provide initial experiments for in Section 4. In our discussion section, we point to three other directions: one is user modeling (term used in statistics and surveys), another is improving LLM-as-a-judge and feedback models, and the third is for the creation of more realistic synthetic data. We point the reviewer to the third paragraph of Appendix B for more details on each of these.
> > >
> > > We hope our response addressed your concerns and questions and we would really appreciate it if you could increase your score!

---

> > > ### Comment · Reviewer_ep5V · 2025-11-23
> > >
> > > Thank you to the authors for their time and effort in responding. However, I did not find additional evidence and am not fully convinced by the rationale behind the evaluation and the value of the proposed user simulation. Therefore, I will maintain my original score.

---

> > > > ### Author Response · Authors · 2025-11-25
> > > >
> > > > Dear reviewer ep5V,
> > > >
> > > > Thank you for your response, we appreciate your time!
> > > >
> > > > It seems like your concern with our work is not the methodology and findings, but **whether more realistic user simulation is needed at all** (“[...] and the value of the proposed user simulation.”). As authors of this submission, we are biased in believing in user simulation as an important research area. However, we point to other researchers in CS beyond our related work that have expressed why they believe research on user simulation is important:
> > > >
> > > >
> > > >  - “User simulators bridge RL with real-world interaction” (https://jessylin.com/2025/07/10/user-simulators-1/)
> > > >
> > > >
> > > >  - “What does it take to build a human-like user simulator?” (https://nickatomlin.github.io/blog/user-simulators-2.html)
> > > >
> > > >
> > > > - A tutorial at AAAI 2024 which points to the “urgent challenge” of building more interactive evaluation of AI systems. (https://usersim.ai/aaai2024-tutorial/)
> > > >
> > > >
> > > > - A recent TechCrunch article describing “simulation environments” as a critical next step to train AI agents. (https://techcrunch.com/2025/09/21/silicon-valley-bets-big-on-environments-to-train-ai-agents/)
> > > >
> > > > We would also like to point to a metaphor: With any new technology (let’s imagine the creation of the first smartphone) it is challenging to know a-priori all the uses that the technology will have, and to propose the critique that the available technology (flip phones) is sufficient. Yet new uses for the technology emerge as it matures. Similarly in our work, we have shown that the most common method to simulate users (prompting an assistant LLM) is deficient and show a method to train more realistic user simulators, which we demonstrate experimentally. Though we cannot experiment with all the possible uses for user simulators in a single paper, we believe that as the field of realistic user simulation grows, uses will expand from the few we have listed in our discussion section.
> > > >
> > > > If your concern is that simply prompting an assistant model is satisfactory for user simulation, we would like to propose a simple practical experiment and invite you to try the following prompt in ChatGPT (or any other platform serving an assistant LM):
> > > >
> > > > *“You must act as a real user (not an assistant) and make requests as a real user would to an AI assistant. You should be brief and potentially make typos like a real user, and not act like an assistant. Your next response should be the user’s first utterance. The task you are trying to accomplish as a user is: getting information about strategies for losing weight and the impact of certain drugs or medications on weight gain.”*
> > > >
> > > > When we do this (in November 2025), GPT-5 generates:
> > > >
> > > > *“uh hey — tryna lose a bit of weight and not sure where to start… also like, do some meds make u gain weight??”*
> > > >
> > > > Though this does follow the instruction of including typos and being unlike “assistant text”, it is excessive and unrealistic. In contrast, in the WildChat conversation for this task, the user’s first utterance was: *“does allertine cause weight gain”*. The second utterance: *“do antihistamines cause weight gain”* and the third utterance: *“how much weight can i lose by cutting sugar for 40 days”*
> > > >
> > > > We hope this clarifies why simply relying on prompted assistants (even with prompts optimizations requesting them to “talk like users”) do not yield realistic user simulators.
> > > >
> > > > We hope the shared artifacts and the practical thought experiment help give some perspective as to why user simulation is relevant and worth pursuing as a research direction!

---

### Official Review · Reviewer_vfFV · 2025-10-31

**Soundness:** 3
**Presentation:** 4
**Contribution:** 3
**Rating:** 8
**Confidence:** 4

**Summary:**

The paper presents 1B and 8B models fine-tuned on the WildChat dataset, where the user utterances were unmasked and the user intent was put as a system message. These models can be used for simulating user utterances in multi-turn conversations.

Other details:
- The intent was inferred with GPT-4o in a few-shot mode.
- The base models were  Llama3-8b-Base and Llama3.2-1b-Base. Starting from the instruction models is worse.
- It was full fine-tuning on 4xA6000.

The main evaluation was by perplexity on the test set (WildChat) and on the OOD test set (PRISM)

Other metrics: first-turn diversity, intent decomposition across turns, dialogue termination, naturalness (Pangram AI-detector), and robustness (role adherence and intent adherence).

User LMs outperform prompted assistant simulators (including GPT‑4o roleplay) and a fine-tuned baseline (USP‑8b), with large gaps in metrics.

Extrinsic evaluations simulate math and coding tasks where UserLM‑8b yields more realistic, varied user behavior and reduces downstream assistant (GPT‑4o) task success from 75% to 57%, suggesting current assistants struggle more under realistic multi-turn conditions.

The authors claim the model will be publicly available for research purposes.

**Strengths:**

The topic is very timely: LLM-based user simulators are needed for offline evaluation of chatbot applications everywhere.

The implementation is very straightforward, which is a good thing in that case, and it is easily reproducible. Other works do almost the same (USP), but this work seems to be both better and simpler in terms of the general design.

The paper presents strong empirical evidence that assistant LMs are poor user simulators; User LMs substantially reduce PPL on PRISM and WildChat and improve multi-turn metrics. Some metric choices are nice, for instance, using the Pangram for naturalness. Conclusions are helpful: base models are better than instruct models; 8B is better than 1B, so scaling works in that case.

The presentation is wonderful, all tables and figures are clean, readable, and well-thought-out.

**Weaknesses:**

1. Extrinsic simulations introduce guardrails applied only to UserLM‑8b (described in Appendix D.1), which gives unfair advantage relative to GPT-based simulators; a fairness ablation is missing.

2. The task scope for extrinsic simulations is very narrow (math/coding) and English-only; generalization to other domains, modalities, and languages is not evaluated. In general, math and code seem like a strange choice.

**Questions:**

1. I am surprised to see USP-8b performing that badly, being worse than Llama3.2-1b-Instruct by PPL. It also seems that USP has more sophisticated training procedures. Why do you think it is that bad?

2. Do guardrails described in Appendix D.1 affect Table 1 results? Or are these guardrails only for the extrinsic simulations?

**Details Of Ethics Concerns:**

The authors make a user simulator that works against the AI detector. It might be harmful.

---

> ### Author Response · Authors · 2025-11-17
> **Response to Reviewer vfFV**
>
> Thank you so much for your efforts reviewing our paper and for recognizing the value of our work with a score of 8, we really appreciate it! We respond to your concerns and questions below.
>
> **(Q1) Use of guardrails in extrinsic evaluation:** We confirm that the guardrails were only used in Section 4, and only affect Table 3 (and not Table 1 and 2). We will clarify this in the paper. We agree that this is indeed an addition that reduces the balance of that experiment. On the other hand, we compare to larger instruct models (GPT-4o-mini and GPT-4o) that have received much more training for instruction following and are more robust with respect to prompting. Without guardrails, we observed that a larger proportion of conversations with UserLM would end prematurely or repeat indefinitely, which reduced the quality of simulation and viability of our experiments. This is indeed a limitation of our UserLM given it is small (8b parameters). We transparently reported the use of such generation guardrails so they can be reused, and we envision that larger, more performance UserLMs in the future, that are trained on more simulation data, will likely see reduced need for these guardrails.
>
> **(Q2) Scope of extrinsic evaluation:** We agree that the scope of our extrinsic evaluation in Section 4 is restricted to only math and code. However, we note that our earlier intrinsic evaluations were done on WildChat and PRISM conversations which are open-ended and represent a variety of domains. We hope that future data releases for domains beyond code and math can help perform extrinsic evaluations beyond those two tasks.
>
> **(Q3) Regarding the USP-8b performance:** We believe that the poor PPL results by USP is due to four factors: (1) USP was trained on a lower volume of data (around 80k conversations), (2) USP was trained on utterances from LMSYS which might have a different user population than WildChat / PRISM, (3) USP was not trained with task intent conditioning, (4) USP used an RL method after finetuning (SFT) that we did not use (and might have led to a loss in generalization).
>
> **(Q4) Regarding the potential need for ethics review:** We agree with the observation of the review that user simulators could have risky uses. We have completed a comprehensive Ethics Review at our organization prior to the submission of the work to ICLR and have conducted an analysis of the potential risks of harmful uses of UserLM, which we are willing to share.
>
> Thank you so much again for reviewing our paper and for your feedback!

---

### Official Review · Reviewer_xhbu · 2025-10-31

**Soundness:** 2
**Presentation:** 3
**Contribution:** 2
**Rating:** 6
**Confidence:** 4

**Summary:**

The paper explores an alternative perspective in large language model (LLM) training by focusing on the user simulator rather than the dialogue model itself. To address the challenge of maintaining user intent across multi-turn conversations, the authors propose an intent extraction and training method. Experimental results on multiple datasets (such as WildChat and PRISM) show consistent improvements over baselines and better align with human behavior by measuring multiple metrics.

**Strengths:**

1. The authors focus on a critical yet underexplored task (user model construction) and propose effective methods that achieve promising results.
2. The authors provide comprehensive and fine-grained analyses, offering valuable insights for future user simulator research.
3. The authors introduce diverse evaluation metrics tailored to user simulators, which can serve as useful references for subsequent evaluation frameworks.

**Weaknesses:**

1. A major concern is that the “FLIPPING the Dialogue” paradigm has already been partially explored in prior works (e.g., USP), which weakens the originality claim. The paper should clearly articulate how it differs from existing approaches.
2. The correctness of extracted intents used for training has not been validated, potentially introducing bias in both model performance and evaluation (e.g., UserLM is trained and tested on WildChat).
3. The baseline comparison is limited since only USP is included, while prior role-playing and persona-based approaches are not discussed. As a result, it is unclear whether the observed gains stem from the proposed methodology itself or simply from incorporating user profiles (in whatever form). Moreover, several metrics (e.g., Table 3) are model-specific, lacking objective comparability.
4. The evaluation relies heavily on LLM-based automatic assessments. Some degree of human evaluation should be included to validate the reliability of the results.

**Questions:**

1.	How exactly is the <|endconversation|> mechanism implemented across models? As mentioned in the paper, is it merely introduced through prompts? For UserLM, was this mechanism explicitly annotated during training? The notable improvements in Table 2 suggest further clarification.
2.	The paper claims that abstract rather than specific intent extraction is designed to avoid simple memorization by LLMs, yet this abstraction could harm intent fidelity. How is the correctness of extracted intents measured or validated?
3.	Are the datasets used for evaluation in Section 3 the same as those introduced in Section 2?
4.	From a writing perspective, the introduction mentions that only two prior works exist in this area (two citations). However, the related work section shows that many studies are indeed relevant. These should be cited earlier, along with a clear positioning of this paper’s novelty. Furthermore, while the analysis is extensive, it lacks a clear structure and logical flow; reorganizing the sections would greatly improve readability.

---

> ### Author Response · Authors · 2025-11-17
> **Response to Reviewer xhbu 1/2**
>
> Thank you so much for your efforts reviewing our paper, we really appreciate it! We respond to your concerns and questions below.
>
> **(W1) Findings not available in other user simulation approaches:** We clarify that our UserLMs provides a more realistic simulation environment compared to two main types of prior work on user simulation:
>
> - The first (and larger) one is prior work leveraging prompted assistant models to simulate users. Our evaluations indicate that such methods are not satisfactory, with the simulators lacking ability to align with the distribution of utterances (Table 1) and not exhibiting important user-like properties (Table 2). We build a smaller specialized model we show outperforms the prompting of larger assistant models. Most importantly, we find that stronger assistants are not necessarily better user simulators, indicating that improved assistants will not lead to strong user simulators. For example, our UserLM-8b was better at simulating realistic user conversations than prompting GPT-4o to simulate a user.
>
> - The second body of work has trained user simulators primarily for the creation of synthetic corpora used to train assistant models (such as the USP-8b model we compare to). In other words, the user simulators they trained were more of a byproduct for another objective (generating synthetic conversations to train assistants), whereas we focus on building a simulator that is steerable towards a particular task. This required us to set up intrinsic evaluation (Section 3) tying desirable properties to quantitative metrics, allowing us to evaluate progress on the task of user simulation itself. In the discussion section (Appendix B), we discuss the potential uses of user simulators beyond evaluation of assistant models, they include: user modeling (in statistics & survey studies), LLM-as-a-judge, and synthetic data generation.
>
>
> **(W2) Validity of generated task intents:** We agree that misalignment between generated task intents and underlying conversations could lead to undesirable behavior, such as lowering the steerability of the user simulator. In our design choices for generating task intents, we chose to make the task intents more generic (i.e., high-level) so that the user simulator would have a sense of the high-level task objectives, but not the exact wording details, which it would have to fill in. We believe this choice also alleviates risks of hallucination/discrepancies in the generated task intents: since the intents only cover the task in conversation without its details, there are fewer elements that could be wrongly included in the task intents. We used one of the most capable models at the time to generate task intents (GPT-4o) for all Wildchat conversations using few-shot prompting with hand-written demonstrations.
>
> **(W3) Baseline comparisons:** We agree that our primary comparison to finetuned models is with USP-8b, since it was the closest and most relevant publicly available model we could find. However, experiments include comparisons to a much larger body of work: prompted assistant models as simulators. In fact, we used prompts from existing related work, which we further engineered (based on validation perplexity), this therefore represents a second class of strong baseline (since it represents how prior work has simulated users). Though these details are in Appendix E, they are not visible and do not present included prompted assistants as a strong baseline. We will add the details on the prompts and their engineering to Section 2, to further clarify the relevance of the comparison.
>
> **(W4) Human validation:** We believe that conducting a user study to evaluate user simulators is challenging. For example, presenting conversations to a reader and asking them to judge the realistic-ness is challenging, as judging human-ness is likely subjective. We however believe that we can collect user utterances for *held out users* that have had conversations with an assistant, and observe to what extent our user simulator is distributionally aligned with such a user. This is in fact what we do in our experiment with the held-out PRISM dataset: we observe that we are more able to predict the utterances of these held out participants than other methods, and therefore this participant population validates our user simulator. We therefore view that one potential way to continue validating user simulators is to hire participants to have conversations with chatbots, and study the distributional alignment between the user simulator and these new participants. We would like to add this framing to our discussion section.

---

> > ### Author Response · Authors · 2025-11-17
> > **Response to Reviewer xhbu 2/2**
> >
> > **(Q1) Implementation of <|endconversation|>:** For UserLM, the <|endconversation|> is added as a special token to the vocabulary (using one of the unused special tokens from the base model’s vocabulary), and then trained on (based on when conversations end in WildChat). For other user simulators, since they are not trained to explicitly predict end of conversation, the prediction is achieved through prompting by adding to the prompt an instruction to produce the exact string “<|endconversation|>” when the user simulator believes the user would end the conversation. This is indeed a method discrepancy, but it is required since prior methods did not have the built-in capability to predict the end of conversation, which we frame as an important property of a user simulator.
> >
> > **(Q2) Generic intents and the risk of hallucination:** This is indeed a correct observation: as the user simulator receives less specific information in the task intents (such as exact medicine names in the exemplar Figure 2), it has to fill in more of the detail, at the risk of hallucinating details that were not intended. We however believe this is inherently controllable by the researcher using the user simulator: details on which the user simulator should not deviate from must be included in the task intent, and details for which deviation is acceptable can be omitted. To give a concrete example, in simulated conversations shown in Appendix Figure 7, the UserLM “hallucinates” that sorting should be implemented with merge sort in one of the simulations, when the sorting algorithm was not specified in the original task intent. This is however acceptable, as it does not affect the validity of the conversation (it simply adds an additional constraint). In short, the generic intent allows the researcher to define what elements in the instruction need to be fixed (and added to the task intent) and other details will have potential deviations, allowing for more diverse simulations.
> >
> > **(Q3) Datasets used:** We clarify that for our evaluations in Section 3, to evaluate intent decomposition and dialogue termination, we used the PRISM dataset that was used in Section 2. For the role-adherence and intent-adherence evaluations of Section 3, we used the CommonsenseQA and NaturalQuestions datasets. We will add these important details to the description of experiments.
> >
> > **(Q4) Related Work Position:** Thank you for your suggestion regarding the organization of the work. We propose to move the Related Work section earlier in the work (prior to the introduction of the method) so that it can serve as a more extensive review of prior work before introducing our contributions.
> >
> > We hope our response addressed your concerns and questions and we would really appreciate it if you could increase your score!

---

> > ### Comment · Reviewer_xhbu · 2025-11-27
> >
> > Thank you for the detailed response. Several concerns have been addressed; however, two issues remain where my perspective differs from the authors.
> >
> > First, regarding the core contribution of building a user simulator: the authors argue that their simulator provides a “more realistic” environment, but the definition of more realistic remains unclear. While I agree that trained models generally outperform prompt-engineered ones, this has already been demonstrated in prior profile-guided trainable simulators such as USP. This paper can therefore be viewed as an extension that validates similar findings in the setting of intent-guided user simulation. The second part of this issue concerns the claim that the other difference is “we focus on building a simulator that is steerable towards a particular task.” I believe this aspect is somewhat under-emphasized. The authors generate user intents with GPT-4o to guide the simulator, ensuring goal-directed behavior under relatively broad constraints. If the authors intend to highlight task-oriented simulation as a key novelty, the evaluation should more explicitly revolve around task-specific metrics or human-likeness under task demands. Although Tables 1, 2, and 3 include task-related metrics, they are not sufficiently targeted to convincingly support this claim. Therefore, I believe the contribution in this area does exist, but to a limited extent.
> >
> > Second, regarding the reliability of user evaluation: my original suggestion for human assessment stems from two concerns. (1) The reliability of the automatically generated intents themselves: if GPT-4o generates incorrect or inconsistent intents, subsequent evaluations may be biased. (2) The reliability of the user simulator: the authors should envision realistic use-case scenarios where a researcher might rely on such a simulator, and design third-party evaluations that are decoupled from the simulator’s own assumptions. Otherwise, the current evaluation, consisting mostly of general metrics, cannot convincingly demonstrate that higher scores truly reflect a better user simulator. (This concern was also raised by other reviewers.)
> >
> > Given these two remaining issues, I understand the authors’ position and constraints. For these reasons, I believe the score I assigned is already appropriate.

---

### Author Response · Authors · 2025-12-02
**Summary for Area Chair**

Dear AC, thank you for your effort in reviewing our submission! Sitting at an average of 4.5, we wanted to give you our perspective on the submission and discussion period. In short, the review scores were divided. We have two champions (xhbu, vfFV) that gave positive scores of 6 and 8, citing excitement about the subject of user simulation, the effectiveness of our method, and the comprehensiveness of our analyses and evaluations. We also had two critical reviewers (ep5V, YAMu) that gave scores of 2.

For the critical reviewers:


- Reviewer ep5V did not express issues with our methods, but rather a more general lack of interest in the research direction of user simulation, citing *“not fully convinced by the rationale behind the evaluation and the value of the proposed user simulation”*. Though it is challenging to argue for an entire research direction (the need for realistic user simulation), we have provided many clarifications and cited further external evidence to reviewer ep5V that highlight the importance of the proposed user simulation, but have unfortunately received little engagement with our rebuttal.



- Reviewer YAMu had concerns regarding the deduplication of the Wildchat training dataset which we provided further evidence and clarification for, with the reviewer acknowledging we addressed their concerns. A second concern was around the extrinsic evaluation we proposed in Section 5: we believe the reviewer had a misunderstanding regarding our extrinsic evaluation and whether the comparison was fair (*“Is UserLM fluent in language usage? How about the difficulty of the tasks requested?”*). We would have liked to clarify that all baselines were tested on the same set of math and coding tasks that are used widely in prior work, making task difficulty fixed by definition. The manner in which the simulators will engage with the assistant to solve those tasks is what changes in our setting, and which we provide further fine-grained analyses for to make our conclusions. Unfortunately, given the circumstances of the review process, we couldn’t continue our discussion with reviewer YAMu.

We imagine submission decisions are challenging, and we appreciate your decision! Thank you for your time and service!

---

### Meta-Review · Area_Chair_3fCs · 2026-01-13

**Summary:**

This paper proposed to model user turns using LMs instead of the prevalent approach of assistant turn modeling. This work finds that existing assistant LMs are poor user simulators and built user LMs to simulate human users. Experiments found their proposed user LMs to align better with user behavior, and that when using them to simulate user turns leads to drops in gpt-4o performance in responding.

Overall, reviewers are very polarized in the evaluation of this work even after rebuttal, and both sides are well justified. Two reviewers provide strongly negative reviews and two reviewer provide positive reviews (one of which being very positive). Given that developing realistic user simulators seems timely and relevant for multi-turn evaluation of assistants, I am recommending acceptance, but this decision can be bumped down given the unaddressed reviewer concerns (detailed below).

**Reviewer Concerns:**

The main concerns are: first, whether the task of user simulation makes sense as unlike assistant simulation with a clear application, user turn modeling is not directly/obviously useful. Second, evaluation seems insufficient as the drop in assistant LM response quality when prompted with generated user turns might be due to other reasons such as quirks in generations, and it seems more direct to do human evaluation to evaluate the generation quality (such as how indistinguishable generations are from real human turns). Third, evaluation scope is limited to math and coding.

**Reviewer Scores:**

The review scores are very polarized, and it seems that the two reviewers who provided strong negative scores are very unlikely to increase their scores: one explicitly stated that they would maintain their score and another replied once without mentioning changing their score. The positive reviewers also seem likely to keep their scores.

---

### Decision · Program_Chairs · 2026-01-26

Accept (Poster)